# Regioselective stilbene *O*-methylations in Saccharinae grasses

Andy C. W. Lui [1,9,10], Kah Chee Pow [2,10], Nan Lin [1,10], Lydia Pui Ying Lam [3], Guoquan Liu [4], Ian D. Godwin [4], Zhuming Fan [2], Chen Jing Khoo [2], Yuki Tobimatsu [5], Lanxiang Wang [6], Quan Hao [2,7,8] ✉ & Clive Lo [1] ✉

*O*-Methylated stilbenes are prominent nutraceuticals but rarely produced by crops. Here, the inherent ability of two Saccharinae grasses to produce regioselectively *O*-methylated stilbenes is reported. A stilbene *O*-methyl-transferase, SbSOMT, is first shown to be indispensable for pathogen-inducible pterostilbene (3,5-bis-*O*-methylated) biosynthesis in sorghum (*Sorghum bicolor*). Phylogenetic analysis indicates the recruitment of genus-specific SOMTs from canonical caffeic acid *O*-methyltransferases (COMTs) after the divergence of *Sorghum* spp. from *Saccharum* spp. In recombinant enzyme assays, SbSOMT and COMTs regioselectively catalyze *O*-methylation of stilbene A-ring and B-ring respectively. Subsequently, SOMT-stilbene crystal structures are presented. Whilst SbSOMT shows global structural resemblance to SbCOMT, molecular characterizations illustrate two hydrophobic residues (Ile144/Phe337) crucial for substrate binding orientation leading to 3,5-bis-*O*-methylations in the A-ring. In contrast, the equivalent residues (Asn128/Asn323) in SbCOMT facilitate an opposite orientation that favors 3′-*O*-methylation in the B-ring. Consistently, a highly-conserved COMT is likely involved in isorhapontigenin (3′-*O*-methylated) formation in wounded wild sugarcane (*Saccharum spontaneum*). Altogether, our work reveals the potential of Saccharinae grasses as a source of *O*-methylated stilbenes, and rationalize the regioselectivity of SOMT activities for bioengineering of *O*-methylated stilbenes.

Stilbenes are specialized metabolites sporadically distributed in the plant kingdom. Grape (*Vitis vinifera*, dicot), peanut (*Arachis hypogaea*, dicot), Scots pine (*Pinus sylvestris*, gymnosperm), sorghum (*Sorghum bicolor*, monocot) and sugarcane (*Saccharum* spp., monocot) are iconic species producing various stilbene compounds upon (a)biotic challenges[1-5]. They serve to maintain reactive oxygen species homeostasis and/or protect against microbial attacks[3,6-8]. Recently, stilbenes have emerged as prominent candidates for nutritional and

[1]School of Biological Sciences, The University of Hong Kong, Pokfulam, Hong Kong, China. [2]School of Biomedical Sciences, LKS Faculty of Medicine, The University of Hong Kong, Pokfulam, Hong Kong, China. [3]Center for Crossover Education, Graduate School of Engineering Science, Akita University, Tegata Gakuen-machi 1-1, Akita City, Akita 010-8502, Japan. [4]Queensland Alliance for Agriculture and Food Innovation, The University of Queensland, Brisbane, QLD 4072, Australia. [5]Research Institute for Sustainable Humanosphere, Kyoto University, Gokasho, Uji, Kyoto 611-0011, Japan. [6]CAS Key Laboratory of Quantitative Engineering Biology, Shenzhen Institute of Synthetic Biology, Shenzhen Institute of Advanced Technology, Chinese Academy of Sciences, Shenzhen 518055, China. [7]Institute of High Energy Physics, Chinese Academy of Sciences, Beijing 100049, China. [8]China Spallation Neutron Source, Dongguan, Guangdong 523000, China. [9]Present address: Plant Breeding and Genetics Section, School of Integrative Plant Science, Cornell University, Ithaca, NY 14853, USA. [10]These authors contributed equally: Andy C.W. Lui, Kah Chee Pow, Nan Lin. ✉e-mail: qhao@hku.hk; clivelo@hku.hk

pharmaceutical studies due to their anti-aging, anti-neurodegeneration, anti-diabetes and chemo-prevention properties[9–11]. Over 200 stilbene-related clinical trials (http://clinicaltrials.gov/) have been launched and stilbenes like resveratrol and pterostilbene are commercialized as dietary supplements. Despite the heightened interest in this minor, yet unique, class of phenylpropanoids as pharmaceuticals and nutraceuticals, *in planta* stilbene biosynthesis remains understudied, especially the downstream derivatization steps such as *O*-methylations and prenylations. Elucidation of the underlying enzymology would expedite bioengineering attempts to fortify crops with more or novel stilbenes.

*O*-Methyltransferases (OMTs) are indispensable in a diversity of metabolic pathways and hence biophysiological processes. For instance, sorghum SbOMT3 catalyzes the biosynthesis of sorgoleones which are allelochemicals exuded by root hairs to suppress competing weeds[12]. Two maize OMTs regioselectively methylate flavonoid phytoalexins which inhibit *Fusarium* spp. infection[13]. The degree of *O*-methylations of anthocyanidin pigments affects floral and fruit colors in flowering plants[14,15]. In Arabidopsis (*Arabidopsis thaliana*), disruption of a tapetum-specific OMT (AtTSM1) which methylates polyamine conjugates adversely affects silique development and seed setting[16]. Additionally, the degree of phenylpropanoid *O*-methylation determines lignin composition in tracheophytes[17]. For example, disruption of caffeic acid *O*-methyltransferase (COMT) and/or caffeoyl-CoA *O*-methyltransferase (CCoAOMT) substantially alters lignin content and composition, potentially affecting plant growth and development[18–20]. Biochemical and structural characterization of these phenylpropanoid OMTs have provided insights into their substrate-enzyme interactions and catalytic mechanisms[21–24], hence facilitating genetic manipulations of lignin composition in biomass crops[25,26].

Phylogenetically-unrelated plant species have independently recruited specific OMTs for stilbene *O*-methylation. In response to environmental stresses, grapevine resveratrol OMT (VvROMT) methylates resveratrol to pterostilbene whereas Scots pine pinosylvin OMT (PsPMT2) methylates pinosylvin to pinosylvin monomethyl ether[27,28]. In addition, a root hair-specific sorgoleone *O*-methyltransferase from sorghum, SbOMT3, methylates resveratrol to pterostilbene in vitro or in transgenic models, albeit at poor efficiency[12,29]. Synthetic biology approaches have been initiated to generate *O*-methylated stilbenes of pharmaceutical significance such as pterostilbene and 3′-hydroxypterostilbene[30,31], which are critically acclaimed for their potent cancer-chemopreventive[32] and neuroprotective properties[33]. In fact, their pharmacological properties are superior to the non-*O*-methylated analogs[34,35]. However, the lack of stilbene *O*-methyltransferase (SOMT) structural data greatly limits our understanding of their distinctive substrate specificities, regioselectivities and catalytic efficiencies, impeding bioengineering efforts in the production of high-value and/or novel *O*-methylated stilbenes.

Within the family Poaceae, the *Sorghum* and *Saccharum* genera evolved from a common ancestral grass and belong to the tribe Andropogonae, subtribe Saccharinae. Species from both genera accumulate hydroxylated stilbenes (e.g. resveratrol, piceatannol) and their *O*-glycosides[5,7,36], but *O*-methylated stilbenes have not been reported in these species so far. Here, we report the identification of *O*-methylated stilbenes, pinostilbene (3-*O*-methylated) and pterostilbene (3,5-bis-*O*-methylated) in sorghum, and isorhapontigenin (3′-*O*-methylated) in wild sugarcane (*S. spontaneum*). A stilbene OMT (SbSOMT) catalyzes sequential 3,5-bis-*O*-methylation of resveratrol to pterostilbene whereas a canonical COMT (SsCOMT) likely catalyzes 3′-*O*-methylation of piceatannol into isorhapontigenin in wild sugarcane. We generated sorghum *SbSOMT* CRISPR/Cas9 mutants which were deficient in pinostilbene and pterostilbene, hence establishing the indispensable role of SbSOMT in resveratrol 3,5-bis-*O*-methylation. As indicated by phylogenetic analysis, divergence of *Sorghum* genus from other genera in the Saccharinae subtribe likely predated the

recruitment of SbSOMT from a canonical COMT. By solving the high-resolution crystal structures of SbSOMT complexed with stilbenes, molecular features that differentiate the substrate binding mode and catalytic regioselectivity between SbSOMT and canonical COMTs were unveiled.

## Results

### Accumulation of *O*-methylated stilbenes in pathogen-infected sorghum

The metabolite profiles of wild-type sorghum upon infection of *Colletotrichum sublineola* (Fig. 1a) were first examined. Two sorghum genotypes, BTx623 and SC748-5, susceptible and resistant to *C. sublineola* infection respectively[37,38], were analyzed. Accordingly, four stilbenes including resveratrol, piceid (resveratrol 3-*O*-glucoside), and resveratrol derivatives *O*-methylated at their A-ring: pinostilbene (3-*O*-methylated resveratrol), and pterostilbene (3,5-bis-*O*-methylated resveratrol) were identified in the infected sorghum mesocotyls (Fig. 1b–f). In addition, flavones (apigenin, luteolin, chrysoeriol, tricin; Supplementary Table 1) and 3-deoxyanthocyanidins (orange-red pigments; apigeninidin, luteolinidin, diosmetinidin; Supplementary Table 2), which are known sorghum phytoalexins, were detected[37,38]. Notably, the resistant genotype SC748-5 accumulated approximately 9-fold more pterostilbene than the susceptible genotype BTx623 starting from 48 h post infection (Fig. 1b & Supplementary Table 3). Taken together, stilbene accumulation in SC748-5 was more rapid, and at larger quantities than BTx623 during *C. sublineola* infection.

### Sorghum pathogen-inducible SbSOMT catalyzes pterostilbene biosynthesis

To identify potential OMTs involved in pterostilbene biosynthesis in sorghum, an in silico expression dataset for *Bipolaris sorghicola*-infected sorghum leaves (http://matsui-lab.riken.jp/morokoshi/)[39,40] and our in-house transcriptome dataset for *C. sublineola*-infected sorghum mesocotyls[41] were analyzed. Transcripts of SbOMT1 and SbOMT3, which were reported to methylate resveratrol in vitro and in transgenic plants, were not detected in either dataset (Supplementary Fig. 1a). Meanwhile, *SbCOMT*, which encodes a bona fide canonical COMT[18,21,22], is constitutively expressed in sorghum (Supplementary Fig. 1a).

Furthermore, two putative *OMT* genes, *SbSOMT* (*Sb07g004710*) and *SbOMT4* (*Sb07g004690*), showed pathogen-inducible expression patterns similar to that of *SbSTS1* (*Sb07g004700*) (Supplementary Fig. 1a) encoding stilbene synthase (STS) which generates resveratrol from *p*-coumaroyl-CoA and malonyl-CoAs (Fig. 1a)[5]. Gene expression analyses confirmed their transcriptional upregulation in both sorghum genotypes shortly after infection (Supplementary Fig. 1b–d). In particular, the resistant genotype SC748-5 showed stronger expression for these genes than the susceptible genotype BTx623, consistent with its higher levels of pathogen-induced stilbene accumulation.

Subsequently, recombinant proteins expressed in *E. coli* were purified for in vitro enzyme assays. Both SbSOMT and SbCOMT generated pinostilbene and/or pterostilbene when incubated with resveratrol albeit at significantly different levels. Notably, SbSOMT converted about 60% of resveratrol into pterostilbene whereas SbCOMT displayed poor resveratrol 3,5-bis-*O*-methylation activities (Fig. 2a). Meanwhile, SbOMT4 showed minimal SOMT activities (Supplementary Fig. 2a–c) and is thus unlikely to contribute to pterostilbene production in sorghum. The catalytic activities of SbSOMT and SbCOMT towards hydroxycinnamic acids were also compared. Intriguingly, SbSOMT failed to methylate caffeic acid and 5-hydroxyferulic acid, whereas SbCOMT efficiently converted them to ferulic acid and sinapic acid, respectively (Fig. 2b–c).

Kinetic parameters of SbSOMT and SbCOMT were then determined using pinostilbene as a substrate since SbSOMT rapidly

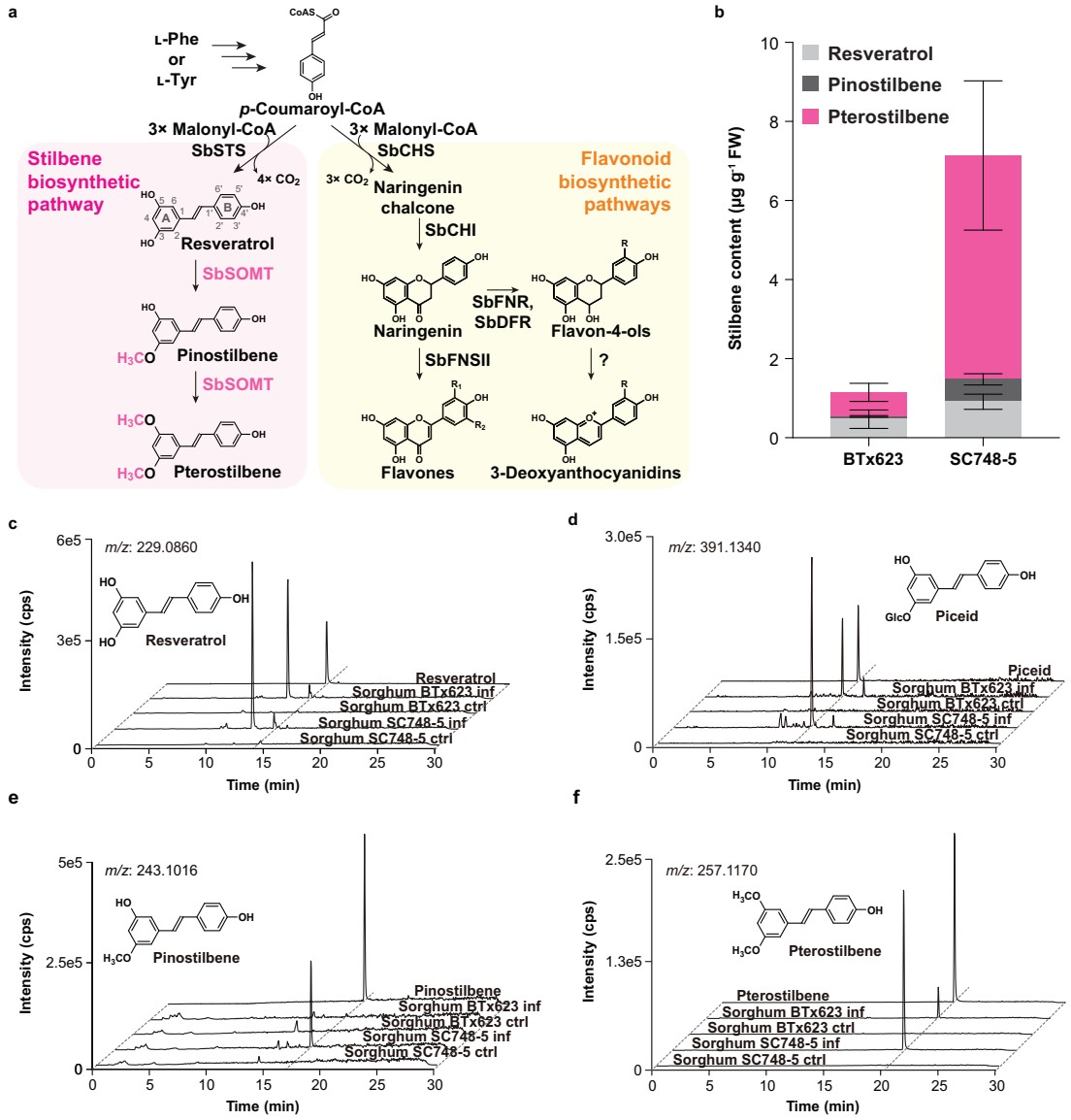

**Fig. 1 | Biosynthesis of stilbenes and flavonoids in sorghum and stilbene profiles of *Colletotrichum sublineola*-infected sorghum mesocotyls. a** Stilbene and flavonoid biosynthetic pathways in sorghum. Newly formed methoxy groups are colored and enlarged. STS stilbene synthase, SOMT stilbene *O*-methyltransferase, COMT caffeic acid *O*-methyltransferase, CHS chalcone synthase, CHI chalcone isomerase, FNR flavanone 4-reductase, DFR dihydroflavonol 4-reductase, FNSII flavone synthase II, Sb *Sorghum bicolor*. R = H, OH or OCH₃. **b** Distribution of stilbene aglycones in *C. sublineola*-infected mesocotyls of sorghum genotypes SC748-5 (resistant) and BTx623 (susceptible) 96 h after treatment. Values refer to means ± SD ($n = 3$). **c–f** HPLC-QTOF-HRMS detection of resveratrol (**c**), piceid (**d**), pinostilbene (**e**) and pterostilbene (**f**) in *C. sublineola*-infected mesocotyls of sorghum genotypes SC748-5 (resistant) and BTx623 (susceptible). inf infected, ctrl control, cps counts per second, Glc glucose.

converts resveratrol into pterostilbene, thus preventing accurate quantification of pinostilbene produced. Results revealed the superior catalytic performance ($k_{cat}/K_m$ and specific activity) of SbSOMT over SbCOMT and SbOMT4 towards pinostilbene, although SbSOMT showed a higher $K_m$ value (Table 1). These data suggested that SbSOMT, but not SbCOMT, is highly likely the primary SOMT for pterostilbene biosynthesis in sorghum.

The biochemical function of SbSOMT was then examined via transient co-overexpression in *Nicotiana benthamiana* (Supplementary Fig. 3a–d). Overexpression of both *SbSTS1* and *SbSOMT* produced pterostilbene as the only stilbene product in agro-infiltrated leaves. By contrast, overexpression of *SbSTS1* alone generated resveratrol and a small amount of pinostilbene, presumably due to endogenous promiscuous activities of tobacco OMT(s). These data are supportive for the role of SbSOMT in 3,5-bis-*O*-methylation of resveratrol to pterostilbene *in planta*.

## Sorghum *SbSOMT* mutants are depleted in pathogen-inducible pinostilbene and pterostilbene

Sorghum *sbsomt* mutants were generated via CRISPR/Cas9-mediated genome editing. Three homozygous *sbsomt* mutant lines (*sbsomt-a*, *sbsomt-b1*, *sbsomt-b2*; T₁ generation) harboring different mutation pattern were isolated for metabolite analysis (Fig. 3a & Supplementary Fig. 4a–c). In all mutant lines, the indels on exon 1 alone were sufficient to induce frameshift mutations and premature translation termination, resulting in knockout mutations (Supplementary Fig. 5).

Metabolite profiles of *sbsomt* mutants were analyzed upon *C. sublineola* infection. (Fig. 3b–e & Supplementary Table 4). Consistent with the results above (Fig. 1c–f), wild-type Tx430 mesocotyls accumulated resveratrol, piceid, pinostilbene and pterostilbene 72 h after infection. By contrast, the *sbsomt-a/b1/b2* mutant mesocotyls accumulated resveratrol and piceid but not pinostilbene or pterostilbene even at 96 h after infection. These results firmly established the

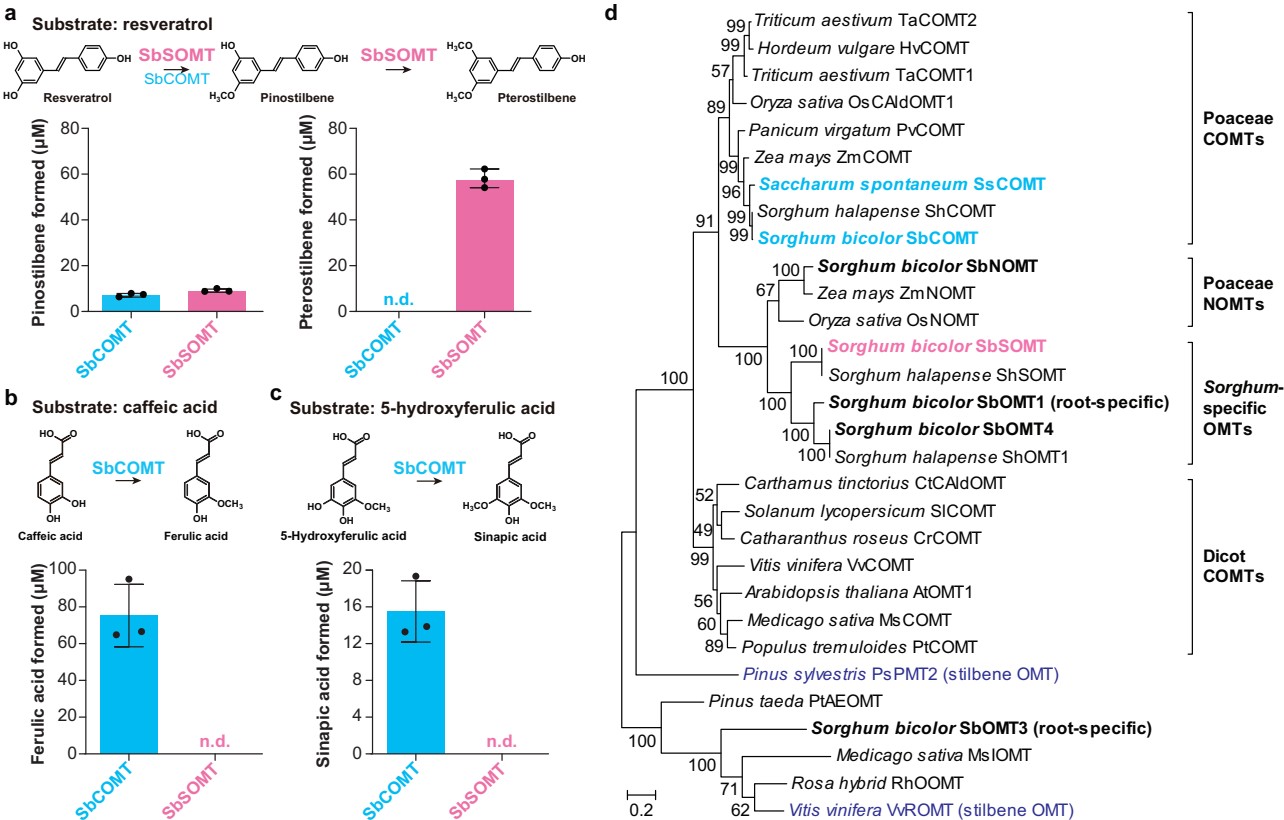

**Fig. 2 | Catalytic activities and phylogeny of SbSOMT and SbCOMT. a** Formation of pinostilbene and pterostilbene from resveratrol by SbSOMT and SbCOMT after two-hour incubation with 100 μM resveratrol. Values refer to means ± SD (*n* = 3). n.d. not detected. Dots represent individual data points. **b** Formation of ferulic acid from caffeic acid by SbSOMT and SbCOMT after two-hour incubation with 100 μM phenylpropanoid substrates. Values refer to means ± SD (*n* = 3). n.d. not detected. Dots represent individual data points. **c** Formation of sinapic acid from

5-hydroxyferulic acid by SbSOMT and SbCOMT after two-hour incubation. Values refer to means ± SD (*n* = 3). n.d. not detected. **d** Phylogenetic analysis of SbSOMT and SbCOMT. The unrooted phylogenetic tree was constructed by maximum likelihood using MEGA X[90]. Bootstrapping with 1000 replications was carried out. Bona fide stilbene *O*-methyltransferases from gymnosperm[27], dicot[28] and monocot (this study) are highlighted. Sorghum OMTs are bolded. Scale bar denotes 0.2 amino acid substitution per site.

indispensable role of SbSOMT in 3,5-bis-*O*-methylation of resveratrol to form pinostilbene and pterostilbene successively in sorghum.

## Sorghum convergently recruited SbSOMT via neofunctionalization of COMT

Previous studies have implicated that SOMTs in grapevine (VvROMT) and Scots pine (PsPMT2) evolved convergently from non-COMT ancestors[27,28]. Multiple sequence alignment showed that SbSOMT shares only 31.9% and 29.9% sequence identity with VvROMT and PsPMT2, respectively (Supplementary Fig. 6). Instead, SbSOMT is phylogenetically related to canonical COMTs in Poaceae (Fig. 2d). In addition, an SbSOMT ortholog (ShSOMT) is found in Johnsongrass (*Sorghum halapense*; Fig. 2d). Meanwhile, no potential SbSOMT orthologs could be retrieved from wild sugarcane (*Saccharum spontaneum*) proteome (https://plants.ensembl.org/; all homologous proteins with <50% identity; Supplementary Fig. 7) although *Sorghum* spp.

and *Saccharum* spp. share a recent common ancestor (both belonging to the same subtribe). Furthermore, ShOMT1, ShSOMT, SbOMT1, SbOMT4, SbSOMT form a *Sorghum*-specific clade sister to Poaceae naringenin 7-*O*-methyltransferases (NOMTs) (Fig. 2d). Hence, these OMTs were most likely recruited via duplication and neofunctionalization of an ancestral grass COMT. Overall, SbSOMT represents an independent and genus-specific acquisition of SOMT activities distinct from those in grapevine and Scots pine.

## Substrate binding affinity and regioselectivity of SbSOMT and SbCOMT

So far, our results revealed that SbSOMT catalyzes the 3,5-bis-*O*-methylation of resveratrol upon fungal infection. Meanwhile, SbCOMT efficiently methylates hydroxycinnamic acids but demonstrates minimal SOMT activities towards resveratrol and pinostilbene which might be attributed to poor substrate binding. However, isothermal titration calorimetry (ITC) study did not corroborate this possibility (Table 2 & Supplementary Fig. 8). In fact, SbCOMT exhibited strong, micromolar binding affinities ($K_d$) towards resveratrol (1.55 μM), pinostilbene (1.79 μM), and pterostilbene (3.99 μM), comparable to those towards hydroxycinnamic acids[21]. Meanwhile, SbSOMT also showed strong binding affinities towards resveratrol and pinostilbene (4.69 μM and 2.86 μM respectively) but a higher $K_d$ to pterostilbene (11.20 μM), demonstrating SbSOMT favors its substrates over product for binding. The SAM binding affinity of SbSOMT was unexpectedly weak ($K_d$ = 90.00 μM), whereas SbCOMT showed a $K_d$ of 13.00 μM which is consistent with a previous study[21]. Collectively, SbSOMT and SbCOMT

### Table 1 | Kinetic parameters of SbSOMT and SbCOMT with pinostilbene as a substrate

| | SbCOMT | SbSOMT |
|---|---|---|
| $K_m$ (μM) | 3.14 ± 1.40 | 4.62 ± 0.46 |
| $k_{cat}$ (min⁻¹) | 0.14 ± 0.01 | 10.30 ± 0.20 |
| $k_{cat}/K_m$ (min⁻¹ μM⁻¹) | 0.045 | 2.229 |
| Specific activity (nmol min⁻¹ mg⁻¹) | 3.46 ± 0.27 | 290.04 ± 5.64 |

Values refer to means ± SD (*n* = 3).

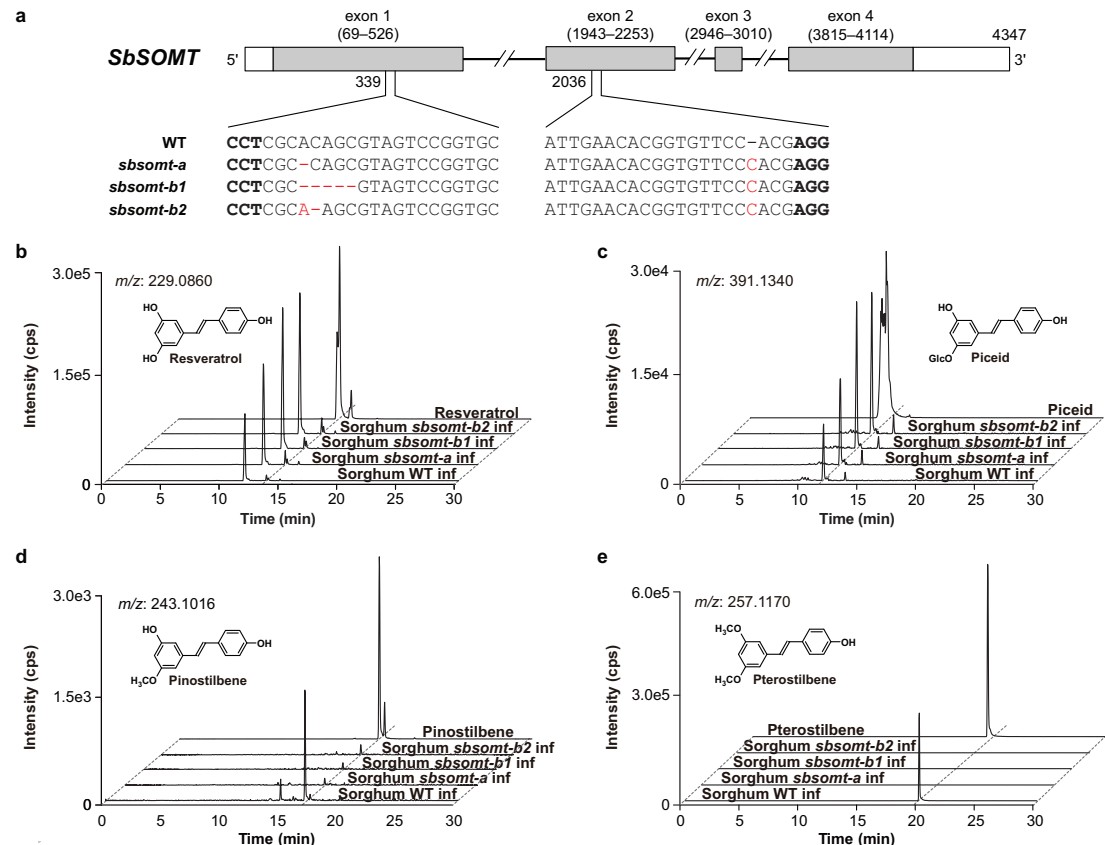

**Fig. 3 | Genotypes, phenotypes and metabolite analysis of sorghum sbsomt mutants. a** Genotypes of sorghum *sbsomt* mutants. In red, deletion or insertion. Bold, protospacer adjacent motif (PAM) site. **b–e**, HPLC-QTOF-HRMS detection of resveratrol (**b**), piceid (**c**), pinostilbene (**d**) and pterostilbene (**e**) in *C. sublineola*-infected mesocotyls of sorghum *sbsomt* mutants after 96 h. inf infected, ctrl control, cps counts per second, Glc glucose.

showed comparable stilbene-binding affinities, which would not account for their different catalytic activities towards resveratrol and pinostilbene.

Canonical COMTs including SbCOMT utilize the *para*-hydroxyl group of phenylpropanoids for proper substrate orientation[21,22] which concurs with our results of SbCOMT-catalyzed *O*-methylation of caffeic acid and 5-hydroxyferulic acid (Fig. 2b). On the other hand, SbSOMT-catalyzed *O*-methylation of resveratrol occurs in the A-ring which is not *para*-hydroxylated. Piceatannol (3′-hydroxylated resveratrol) was then used as an enzyme substrate since its stilbene B-ring is structurally identical to the phenolic ring in caffeic acid while its A-ring is identical to that in resveratrol (Fig. 4a). Both enzymes catalyzed in vitro *O*-methylation of piceatannol, but occurring in a regioselective manner (Fig. 4b–c). SbSOMT methylated the 3- and 5-hydroxyl groups on the piceatannol A-ring to produce 3′-hydroxypinostilbene and 3′-hydroxypterostilbene

### Table 2 | Dissociation constants of SbSOMT, SbCOMT with stilbenes

| | $K_d$ (µM) | |
|---|---|---|
| | **SbCOMT** | **SbSOMT** |
| Resveratrol | 1.55 ± 0.13 | 4.69 ± 0.54 |
| Pinostilbene | 1.79 ± 0.09 | 2.86 ± 0.33 |
| Pterostilbene | 3.99 ± 0.24 | 11.20 ± 2.23 |
| Piceatannol | 1.26 ± 0.06 | 37.50 ± 4.33 |
| SAM | 13.00 ± 0.83 | 90.00 ± 9.59 |

Results are expressed as means of $K_d$ ± standard error derived from curve fitting.

successively. By contrast, SbCOMT converted a substantial amount of piceatannol (*ca.* 60%) into isorhapontigenin by 3′-*O*-methylation in the B-ring with a conversion rate similar to those of SbCOMT-caffeic acid and SbSOMT-resveratrol reactions (Fig. 2a–b). A small amount of isorhapontigenin was further methylated at its A-ring to 3′-methoxypinostilbene by SbCOMT (Fig. 4b–c) and the conversion rate was similar to that of SbCOMT-resveratrol reaction (Fig. 2a). Furthermore, ITC experiments showed that SbCOMT displayed significantly stronger binding affinity towards piceatannol when compared to SbSOMT (Table 2; $K_d$ = 1.26 µM and 37.50 µM, respectively). Overall, these experiments strongly demonstrated the different regioselective *O*-methylation properties of SbSOMT and SbCOMT.

### Global structure of SbSOMT resembles that of SbCOMT

To rationalize their different catalytic regioselectivities, structural analyses involving X-ray crystallography and computational methods were conducted. We solved the crystal structure of an SbSOMT-resveratrol-nicotinamide adenine dinucleotide (β-NAD) ternary complex (diffracted to 1.72 Å resolution, Fig. 4d; β-NAD was utilized as an additive to improve crystallization[42]), an SbSOMT-resveratrol binary complex, and two ternary complexes reproduced with pinostilbene or pterostilbene (diffracted at 2.10 and 2.56 Å, respectively; Supplementary Fig. 9a). All complexes were in high structural resemblance and depicted as a homodimer with an open conformation. Super-imposition of SbSOMT and SbCOMT (PDB: 4PGH[21]) revealed their high structural similarity (RMSD = 2.322 Å), except that SbSOMT harbors an extended loop (Val103-Cys114) that forms a threonine-rich dimerization interface (Fig. 4d).

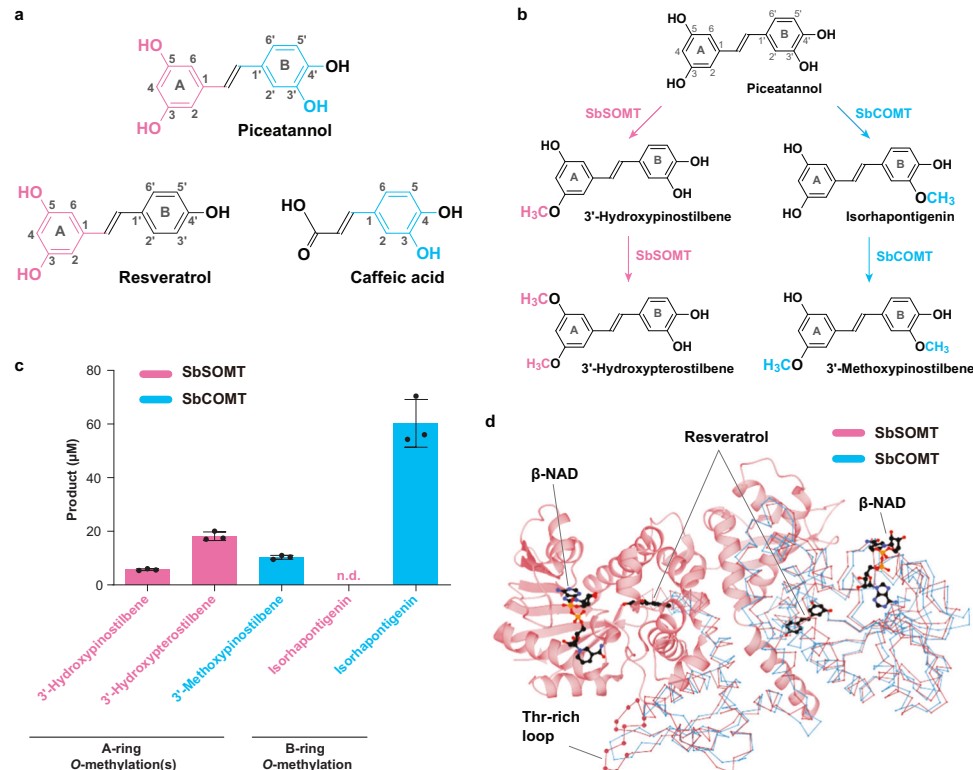

**Fig. 4 | Regioselective *O*-methylation of piceatannol and three-dimensional structure of SbSOMT and SbCOMT. a** Structural comparison between resveratrol, caffeic acid, and piceatannol. Structural similarities are colored accordingly (pink: resveratrol-piceatannol; blue: caffeic acid-piceatannol). **b** Regioselective *O*-methylation of piceatannol catalyzed by SbSOMT (in pink) and SbCOMT (in blue). **c** Formation of different A- and B-ring *O*-methylated products from piceatannol by SbSOMT and SbCOMT after two-hour reaction with 100 μM stilbenes. Values refer to means ± SD (*n* = 3). n.d. not detected. Dots represent individual data points.

**d** Global structure of dimeric SbSOMT-resveratrol-β-NAD ternary complex solved at 1.72 Å resolution. The global structure highlights the position of resveratrol (substrate, black) and β-NAD (additive, black), and the secondary structure of SbSOMT (left protomer, pink). On the right protomer, the chain traces depict the high structural similarity between SbSOMT and SbCOMT (blue; retrieved from PDB: 4PGH;[21] superimposition RMSD = 2.322 Å). The elongated threonine-rich loop (Val103-Cys114) of SbSOMT is highlighted in ball-stick style.

## Structural analyses and in silico docking support different stilbene orientations within the substrate binding pockets of SbSOMT and SbCOMT

Close examination of the SbSOMT substrate binding pocket within the SbSOMT-resveratrol-β-NAD complex provided further insights into the interactions between SbSOMT and its stilbene substrates. The pocket mainly comprised hydrophobic residues, and the stilbene backbone was secured by hydrophobic interactions with Met143, Ile144, Met175, Phe189, Met193, Trp279, Tyr311, Leu329, Met333, Thr336, in addition to Leu28 of the adjacent protomer (Fig. 5a). The apparent binding cavity was confined to fit the stilbene backbone in a planar configuration with minimal rotational flexibility. Of the three hydroxyl groups on resveratrol capable of forming hydrogen bonds, only the methyl-accepting 3-OH group (A-ring) formed direct hydrogen bonds to SbSOMT residues (Fig. 5b), including the catalytic residues His282 (2.80 Å) and Asp283 (3.26 Å). Other hydroxyl groups were shielded by water molecules and did not directly interact with SbSOMT. The same binding mode was adopted for pinostilbene or pterostilbene in the corresponding SbSOMT co-crystals (Fig. 5c–d & Supplementary Fig. 9b). Overall, these structures depicted a productive substrate orientation within the substrate binding pocket which greatly favors A-ring *O*-methylation. Following 3-*O*-methylation of resveratrol, the pinostilbene intermediate will need to be dissociated and re-inserted in order to re-position the A-ring for 5-*O*-methylation to produce pterostilbene. Similarly, in silico docking of piceatannol with SbSOMT (Fig. 5e; in pink) conformed to such substrate orientation, with the A-ring positioned in proximity to the catalytic residues of

SbSOMT, hence favoring the 3,5-bis-*O*-methylation to produce 3′-hydroxypinostilbene and 3′-hydroxypterostilbene (Fig. 4b–c).

Intriguingly, docking stilbenes with SbCOMT (PDB: 4PGH[21]) revealed a contrasting substrate orientation to that of SbSOMT, suggesting that SbCOMT preferentially positions stilbenes with the B-ring close to the catalytic residues and the A-ring reaching inwardly into the pocket. SbCOMT-piceatannol docking showed a single favorable orientation of piceatannol among five hits which were docked at a suboptimal position (Fig. 5e; in blue). In the top hit, the B-ring 3′- and 4′-hydroxyl groups were stabilized by hydrogen bonds with Asn323 while the A-ring 5-hydroxyl group interacted with Asn128 (Fig. 5e; in blue). Consistent with our enzyme assay results (Fig. 2a), the 3′-hydroxyl group on the B-ring was oriented close to the catalytic residues and would thus predominantly favors the methylation of 3′-hydroxyl group (B-ring) over 3-/5-hydroxyl groups (A-ring) (Fig. 5e; in blue). Additionally, docking resveratrol with SbCOMT revealed two opposite orientations with similar free energies of binding (Fig. 5f), representing two competing binding modes which hampered its catalytic performance (Fig. 2a). In fact, the non-productive orientation with the B-ring closer to the catalytic residues was more energetically favorable (−7.3 kcal/mol, in blue), while the productive orientation with A-ring closer to the catalytic residues was slightly less stable (−7.1 kcal/mol, in grey). Collectively, these analyses fully corroborated with results of our biochemical assays and stilbene profiles of sorghum, and provided a mechanistic rationale for the apparent differences in catalytic activity between SbCOMT and SbSOMT. Additionally, as the methyl-accepting -OH groups in piceatannol and

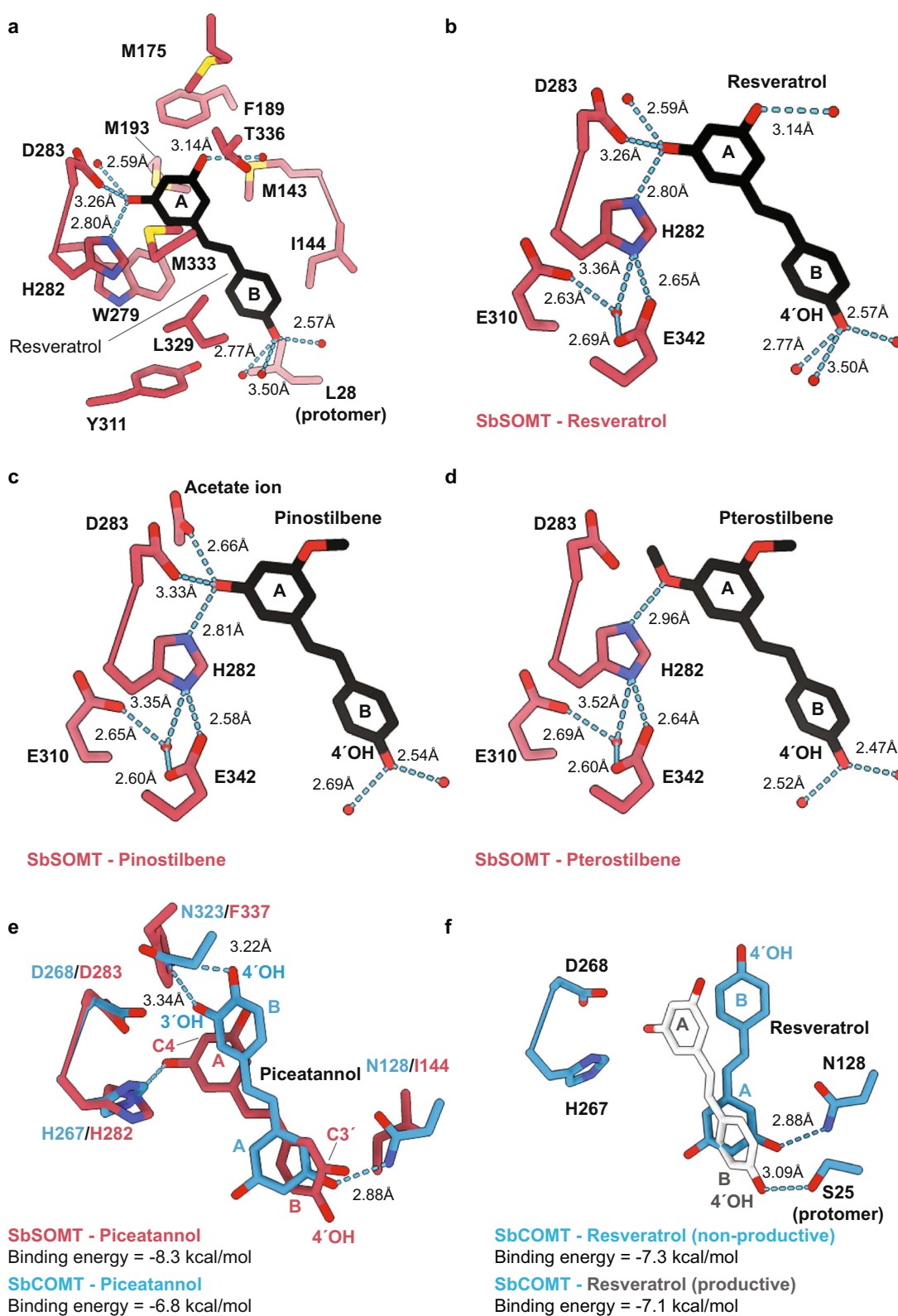

**Fig. 5 | Interactions between stilbenes and SbSOMT catalytic residues, and docking positions of stilbenes with SbSOMT and SbCOMT. a** Close-up view of SbSOMT substrate binding pocket revealed resveratrol conformation and resveratrol-interacting residues and water. Dashed line indicates hydrogen bond formation with the distance labelled. Red dots represent water molecules. **b–d** Close-up view on interactions between SbSOMT catalytic residues (His282, Asp283, Glu310 and Glu342) and resveratrol (**b**), pinostilbene (**c**), and pterostilbene (**d**). Dashed line indicates hydrogen bond formation with the distance labelled. **e** Resulted piceatannol conformation from docking of SbSOMT (pink) and SbCOMT (blue) and relevant interacting residues. Dashed line indicates hydrogen bond formation with the distance labelled. **f** Resulted resveratrol conformation from docking with SbCOMT and relevant interacting residues. Dashed line indicates hydrogen bond formation with the distance labelled.

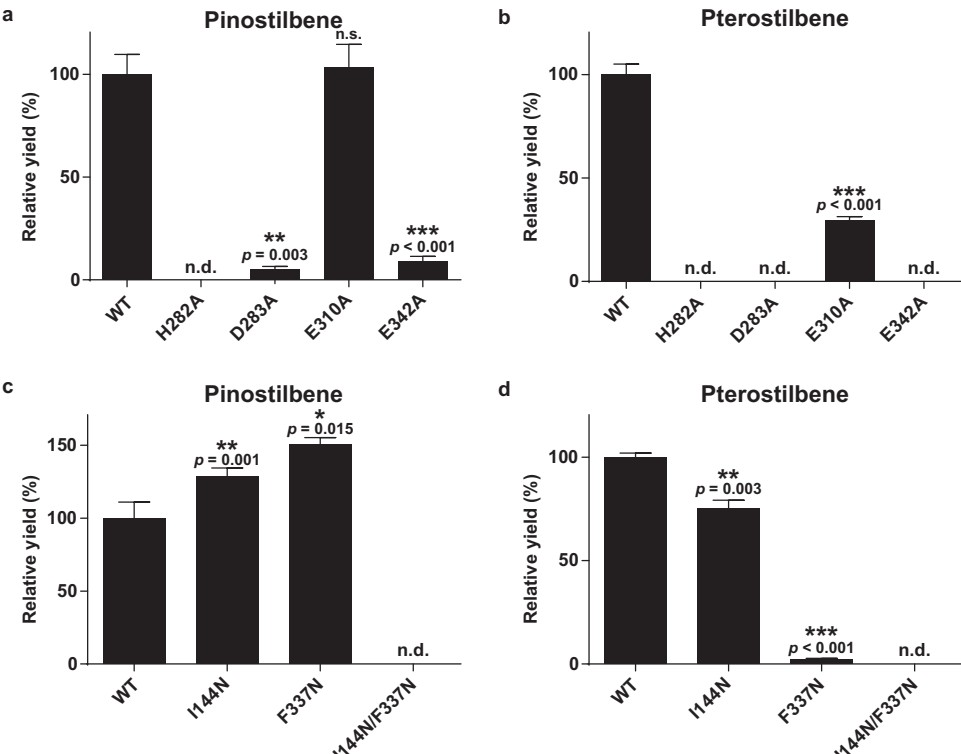

**Fig. 6 | Validation of catalytic residues and key substrate-orientating residues of SbSOMT via site-directed mutagenesis. a–b** Formation of pinostilbene (**a**) and pterostilbene (**b**) by catalytic residue mutant proteins over a one-hour incubation period. Data are expressed as yields relative to that of wild-type SbSOMT protein (WT). **c–d**, Formation of pinostilbene (**c**) and pterostilbene (**d**) by key substrate-orientating residue mutant proteins over a one-hour incubation period. Data are expressed as yields relative to that of wild-type SbSOMT protein (WT). Values refer to means ± SD ($n$ = 3). Asterisks indicate significant differences in yields by mutant protein relative to that of WT (two-sided Student's $t$-test or Welch $t$-test, *$p$ < 0.05, **$p$<0.01, ***$p$ < 0.001 with the exact $p$-values and $t$-test used shown in the Source Data file). Tricin was used as an internal standard for quantitation. Note that pterostilbene represents the major product for SbSOMT(WT)-resveratrol reactions at the 2-hour end point (Fig. 2a). n.d. not detected. Dots represent individual data points.

hydroxycinnamic acids are both adjacent to a *para*-OH group (i.e., 4'-OH in piceatannol and 4-OH in hydroxycinnamic acids), this may be a prerequisite for efficient *O*-methylation by COMTs as suggested previously[21,22].

### SbSOMT and SbCOMT employ the same catalytic residues but different substrate binding orientations

The catalytic residues of SbSOMT are highly conserved with all reported COMTs, including His282 as the key nucleophile, whereas Asp283, Glu310, and Glu342 collectively charge the basicity or confine His282 in its reactive conformation (Fig. 5b)[21,22,43,44]. Accordingly, the surrounding electrostatic force allows Nε of His282 to deprotonate the methyl-accepting -OH group, followed by a $SN_2$ nucleophilic attack of the methyl-donor group on SAM by the oxyanion (-O⁻) (Supplementary Fig. 10)[21,22,44]. The roles of His282 and Asp283 have been validated by mutagenesis, but not for those of Glu310 and Glu342 (PDB: 4PGG[21], 5ICG[45], 1KYZ[44], and 6I5Z[46]). Here, H282A, D283A, E310A, and E342A mutant proteins of SbSOMT were generated for catalytic and binding assays with resveratrol. H282A mutant protein was catalytically defective (Fig. 6a–b) and unstable for ITC study. To further elucidate the role of His282 in resveratrol binding (Fig. 5b), a stable SbSOMT H282N/D283A double mutant protein was generated. While the D283A protein showed a slightly stronger affinity towards resveratrol ($K_d$ = 1.31 μM), the H282N/D283A mutant protein showed a significantly weaker affinity ($K_d$ = 32.90 μM) (Supplementary Table 5), implying that His282 plays a moonlighting role in substrate binding in SbSOMT in addition to catalysis. Meanwhile, D283A and E342A mutations resulted in significant reduction in pinostilbene yield by 94.7% and 91.9%, respectively (Fig. 6a). In addition, both of them failed to

generate pterostilbene when incubated with resveratrol (Fig. 6b) despite that they showed considerably stronger affinities to resveratrol (Supplementary Table 5). Unexpectedly, E310A mutation only caused reduction of pterostilbene yield (by 70.1%; Fig. 6b), accompanying with reshuffled stilbene binding affinities (Supplementary Table 5). We reasoned that the E310A mutation decreased the rigidity and basicity of His282, thus weakening SbSOMT selectivity towards the reactive hydroxyl group against the inactive methoxy group, which is a vital feature allowing the subsequent *O*-methylation of the same stilbene substrate (Fig. 5b–d).

In SbCOMT, two key amino acid residues, Asn128 and Asn323, cooperatively mediate the productive orientation for catalysis[21]. Docking piceatannol with SbCOMT revealed that Asn323 forms a hydrogen bond with the *para*-OH group on the stilbene B-ring whose *meta*-OH group is positioned optimally for methylation, while Asn128 likely serves to stabilize their interaction via another hydrogen bond with 5-OH on the non-*para*-hydroxylated stilbene A-ring (Fig. 5e; in blue). In striking contrast, the equivalent positions in SbSOMT are occupied by two highly hydrophobic residues, Ile144 and Phe337, which are unfavorable for interactions with the *para*-OH group of an aromatic ring as predicted by in silico docking (Fig. 5e; in pink). On the other hand, these residues may contribute to accommodate the stilbene A-ring, which is not *para*-hydroxylated, for methylation of its *meta*-hydroxyl groups. We then generated I144N, F337N and I144N/F337N SbSOMT mutant proteins for catalytic and binding assays. All three SbSOMT mutant proteins showed significantly weakened binding for resveratrol, with F337N and I144N/F337N mutant proteins displaying undetectable binding affinities (Supplementary Table 5). Correspondingly, both I144N and F337N produced substantially less

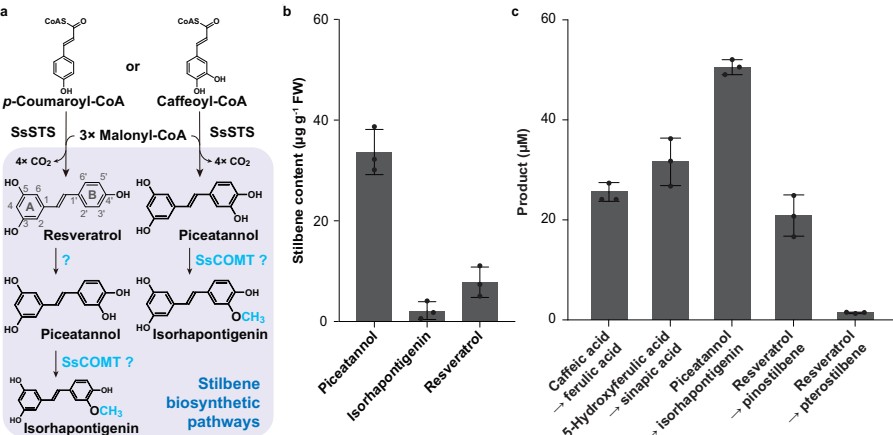

**Fig. 7 | Biosynthesis of stilbenes in wild sugarcane, stilbene profiles of mechanically-wounded wild sugarcane stalks, and in vitro enzyme activities of SsCOMT. a** Proposed stilbene pathways in wild sugarcane. Newly formed methoxy groups are colored and bolded. STS stilbene synthase, COMT caffeic acid *O*-methyltransferase, Ss *Saccharum spontaneum*. **b** Distribution of stilbene aglycones in mechanically-wounded wild sugarcane stalks 120 h after treatment. Values refer to means ± SD ($n = 3$). **c** Formation of ferulic acid, sinapic acid, isorhapontigenin, pinostilbene, and pterostilbene from caffeic acid, 5-hydroxyferulic acid, piceatannol, and resveratrol respectively, by SsCOMT, after two-hour incubation with the indicated substrates (100 μM). Tricin was used as an internal standard for quantitation. Values refer to means ± SD ($n = 3$).

pterostilbene (24.8% and 97.7% respectively), and accumulated slightly more pinostilbene when compared to the WT protein (28.4% and 50.8% respectively; Fig. 6c–d). Meanwhile, the I144N/F337N mutant protein showed complete abolishment of SOMT activities (Fig. 6c–d). As pterostilbene represents the major product generated by SbSOMT-resveratrol reactions (Fig. 2a), these results conclusively established the essential role of Ile144 and Phe337 in the resveratrol binding mechanism adopted by SbSOMT.

## A canonical COMT likely catalyzes isorhapontigenin biosynthesis in wild sugarcane

Previously, *Saccharum* spp. were reported to accumulate resveratrol and piceatannol (3′-hydroxylated resveratrol) upon mechanical wounding or fungal infection[2,36] (Fig. 7a). As *Saccharum* spp. are Saccharinae grasses like *Sorghum* spp., we investigated whether *O*-methylated derivatives could also be detected in wild sugarcane whose genome is available[47]. The slender stalks of flowering wild sugarcane (*S. spontaneum*) were sawn into 2-cm segments (excluding nodes) for metabolite profiling. No stilbenes were detected in freshly-cut segments whereas resveratrol, piceatannol, and isorhapontigenin (3′-*O*-methylated piceatannol) started to accumulate 72 h after wounding (Fig. 7b, Supplementary Fig. 11a–c, and Supplementary Table 6). A miniscule amount of a hexoside of piceatannol was also tentatively identified (Supplementary Fig. 12a–b). Overall, mechanically-wounded wild sugarcane produced piceatannol and its B-ring *O*-methylated derivative isorhapontigenin which constituted a small portion of the stilbene profile (Fig. 7b), while no A-ring *O*-methylated stilbenes could be detected.

Next, we attempted to understand isorhapontigenin biosynthesis in wild sugarcane. Substantial upregulation of *SsSTS* gene expression (*Sspon.06G0010290-2P*) 72 h after wounding was detected (Supplementary Fig. 13a, 13c & 14). Under in vitro condition, recombinant SsSTS generated piceatannol with caffeoyl-CoA as the starter substrate but failed to generate isorhapontigenin when feruloyl-CoA was used (Supplementary Fig. 15a–e). Hence, SsSTS is a functional STS but an OMT is most likely required for 3′-*O*-methylation of piceatannol to produce isorhapontigenin. Meanwhile, a canonical SsCOMT (*Sspon.06g0010980-3C*) highly conserved with SbCOMT (>94.0% protein sequence identity) was identified in wild sugarcane (Supplementary Fig. 7 & 16). It also harbors the key Asn residues corresponding to Asn128 and Asn323 in SbCOMT (Supplementary Fig. 16). In

fact, recombinant enzyme assays demonstrated that SsCOMT and SbCOMT showed the same catalytic regioselectivity. Accordingly, isorhapontigenin, ferulic acid, and sinapic acid were generated when SsCOMT was incubated with piceatannol, caffeic acid, and 5-hydroxyferulic acid respectively (Fig. 7c & Supplementary Fig. 17a–f). However, only small amounts of pterostilbene were generated when resveratrol was used as substrate (Supplementary Fig. 17g–i). In addition, SsCOMT shares similar substrate-binding properties with SbCOMT (Supplementary Table 7). Gene expression analysis further revealed that *SsCOMT* is constitutively expressed in wild sugarcane segments despite its downregulation after wounding (Supplementary Fig. 13b & 13d). Taken together, SsCOMT is a potential candidate for isorhapontigenin production through 3′-*O*-methylation of piceatannol in the B-ring.

## Discussion

*O*-Methylated stilbenes are well-acclaimed for their health-promoting benefits and exceptional bioavailability[35,48]. Sorghum is a major staple crop and wild sugarcane is a genetic resource for sugarcane breeding. In this study, the potential of these Sacharinae grasses as biofactories for *O*-methylated stilbenes as well as the OMTs for regioselective stilbene *O*-methylations were revealed. Our findings also provide insights into bioengineering of specific *O*-methylated stilbenes via molecular breeding and transgenic approaches, representing a unique opportunity to improve dietary intake of these nutraceuticals which are scarcely present in natural food sources.

Recent breakthroughs in sorghum biotechnology, including stable transformation and CRISPR/Cas9-mediated genome editing as well as its metabolic versatility, have warranted sorghum as an emerging model for investigation and manipulation of specialized metabolism[49–51]. Here, complete depletion of *O*-methylated stilbenes in infected sorghum *SbSOMT* CRISPR/Cas9 mutants (Fig. 3a–e) firmly established SbSOMT as the primary SOMT for pathogen-inducible pterostilbene biosynthesis. Concomitantly, molecular, biochemical, and structural characterizations were integrated to elucidate the mechanistic details of SbSOMT-stilbene reactions. We report the first crystal structure of a bona fide stilbene *O*-methyltransferase, SbSOMT. The enzyme utilizes the His282-Asp283-Glu310-Glu342 catalytic residues to generate 3,5-bis-*O*-methylated stilbenes (Figs. 2a, 4b–c, 5b–f, & Supplementary Fig. 9a–c). The same catalytic residues are found in canonical COMTs, indicating the highly conserved *O*-methylation

catalysis irrespective of phenolic substrates involved[13,21,22,52,53]. We further demonstrated that Glu310 likely positions the 5-OH group, instead of the more hydrophobic 3-OCH$_3$ group, in close proximity to His282 for the second O-methylation (Fig. 5c–d & Supplementary Fig. 9b). As Glu310 is highly conserved in most OMTs (with some harboring an Asp), this feature may represent a common mechanism for mediating successive O-methylations of structurally different phenolic substrates.

Meanwhile, our combinatorial analyses rationalized the regioselectivities of SbSOMT and canonical COMTs (e.g., SbCOMT and SsCOMT) underlain by specific substrate binding modes instead of alterations in catalysis or substrate affinity (Fig. 2a, 4b–c, 5b–f, 6a–d, 7a–c, Table 1–2, Supplementary Fig. 10, & Supplementary Table 5). The hydrophilic Asn128/Asn323 residues in SbCOMT, which are highly conserved in COMTs from diverse plant lineages, orchestrate productive coordination with their native substrates, i.e. hydroxycinnamic acids and their analogs, via hydrogen bonding[21,22]. The same binding mechanism is apparently crucial for attaining productive and energetically favorable orientation of piceatannol inside the binding pocket of Saccharinae COMTs including SbCOMT$^{Asn128/Asn323}$ and SsCOMT$^{Asn130/Asn323}$ for B-ring O-methylation (Fig. 5e & Supplementary Fig. 17a–b). In marked comparison, the recruitment of more hydrophobic residues in SbSOMT like Ile144 and Phe337 during its emergence in the Sorghum genus would considerably favor hydrophobic interactions with C3´ and C4 of the stilbene backbone, respectively (Fig. 5e; in pink), hence favoring A-ring O-methylation (Fig. 4b–c, & 6c–d).

Thorough mining of the Protein Data Bank (PDB) retrieved a diverse panel of 24 OMT-ligand complexes (from 13 OMTs with ≥30% protein identity to SbCOMT/SbSOMT), supporting a general occurrence of compatible polarity pairing between the amino acid residue equivalent to SbCOMT$^{Asn323}$/SbSOMT$^{Phe337}$ in OMTs and their substrates (Supplementary Table 8). We reasoned that upon conformational transitions, this residue and the catalytic residues are brought into close proximity, leading to compatible polarity pairing between 12 (out of 13) OMTs and their respective substrate (Supplementary Fig. 18, 19). For example, six OMTs harboring an Asn residue at this position is paired to a polar-OH or -OCH$_3$ moiety of the ligand (Supplementary Table 8 & Supplementary Fig. 19)[43,44,46,54–57]. For the others, the equivalent residue is a more hydrophobic residue (Thr, Val or Phe) which is paired to a non-polar group such as a methyl group or an aromatic ring of the ligand (Supplementary Table 8 & Supplementary Fig. 19). Consistently, two previously characterized SOMTs, VvROMT$^{Phe318}$ and SbOMT3$^{Ile336}$, harbor a hydrophobic residue at this equivalent position[28,29]. Meanwhile, naringenin OMTs, which harbor a hydrophobic Leu residue (OsNOMT;$^{Leu335}$ SbNOMT;$^{Leu324}$ ZmNOMT$^{Leu325}$) paired to the aromatic carbon (C6), catalyze 7-O-methylation (A-ring) of naringenin to generate sakuranetin, but do not utilize hydroxycinnamic acid substrates[58]. Collectively, the polarity of the amino acid residue in OMTs equivalent to SbCOMT$^{Asn323}$/SbSOMT$^{Phe337}$ may govern substrate selectivity through a compatible polarity pairing with the functional group vicinal to the methyl-accepting -OH group in a substrate (Supplementary Fig. 18, 19). Consequently, replacement of Asn323 by a hydrophobic residue likely represents a key molecular event towards the functional divergence of regioselective OMT activities from canonical COMTs in plants. On the other hand, polarity pairing between ligand and SbCOMT$^{Asn128}$ is only observed in some OMT-ligand complexes, as the equivalent residues in OMTs are diversified (Supplementary Table 8). Overall, our findings underpin subsequent bio-engineering of OMTs, either by directed evolution or targeted site-directed mutagenesis, for production of regioselective O-methylated phenolic nutraceuticals/pharmaceuticals in vitro or in planta.

Sorghum and wild sugarcane are closely-related species but resveratrol is their only common stilbene detected by our experimental conditions. In addition to resveratrol, sorghum produces pinostilbene and pterostilbene upon fungal infection (Fig. 1) while wild sugarcane accumulates resveratrol, piceatannol, and isorhapontigenin following mechanical wounding (Fig. 7). The lack of B-ring O-methylated stilbenes (pinostilbene and pterostilbene) in wounded wild sugarcane likely stems primarily from the absence of SbSOMT orthologs which are unique to Sorghum spp. On the other hand, the presence of isorhapontigenin (B-ring O-methylated) is likely resulting from canonical activities of SsCOMT considering its constitutive expression in wounded wild sugarcane stalk (Supplementary Fig. 13b & 13d) and apparent catalytic regioselectivity towards stilbene B-ring (Fig. 7c), although genetic evidence is required for confirmation. Meanwhile, the underlying causes for the exclusivity of piceatannol (3´-hydroxylated) and related stilbenes in Saccharum spp. within the Saccharinae subtribe remain elusive. Elucidation of the piceatannol biosynthetic pathway, in combination with the aforementioned bioengineering of OMTs for targeted stilbene O-methylation, would further unleash the potential of these Saccharinae crops for stilbene biofortification.

Apparently, Sorghum-specific SOMTs was originated from the canonical and ubiquitous COMTs in Poaceae (Fig. 2d). The COMT-to-SOMT evolution fits well into the generally-accepted Yčas-Jensen model which suggests that a promiscuous enzyme with poor activities towards specific substrates often serves as the basic scaffold undergoing sequence divergence (e.g. the aforementioned replacement of SbCOMT$^{Asn323}$ by SbSOMT$^{Phe337}$) to optimize catalytic efficiency[59,60]. Further nonsynonymous mutations within an ancestral sorghum COMT allowed neofunctionalization with acquisition of efficient resveratrol 3,5-bis-O-methylation activities in SbSOMT (Fig. 2a). Such molecular evolution might have been driven by the superior potency of pterostilbene over resveratrol and pinostilbene as a broad-spectrum phytoalexin (Supplementary Fig. 20a–m)[6,61]. Interestingly, both SbSOMT (Fig. 2b) and OsNOMT[58] have lost the ability to O-methylate hydroxycinnamic acids, which are key in planta substrates of COMTs[21,22]. Further investigations will shed new light on whether compromised substrate recognition pattern is necessary, and to what extent, to achieve the highly efficient SOMT activities in SbSOMT. Since both SbSOMT and SbCOMT utilize the same catalytic residues and mechanism for O-methylation, evolution of pterostilbene biosynthesis primarily hinged on optimizing the stilbene binding mode with productive orientations. Furthermore, the recruitment of SbSTS1 from a chalcone synthase scaffold for resveratrol biosynthesis would logically predate that of SbSOMT from SbCOMT[5]. An increasing number of specialized metabolic pathways are considered to have evolved in such order as gene clusters[62–64]. Correspondingly, physically-linked SbSTS1 (Sb07g004700), SbOMT4 (Sb07g004690), and SbSOMT (Sb07g004710) (Supplementary Fig. 21) were identified as constituents of a gene cluster for specialized metabolism (Cluster 408) in the sorghum genome[63,65]. Future elucidation of the regulatory mechanisms and evolution of this gene cluster may unveil the biological and ecological significances underlying the convergent evolution of pterostilbene biosynthesis exclusive to Sorghum spp. among grasses.

## Methods

### Plant materials and fungal treatment conditions

Two sorghum (Sorghum bicolor) genotypes, BTx623 and SC748-5, were analyzed in this study. Sorghum seeds were germinated in aerated double-deionized water at 30 °C for 24 h. Wild sugarcane (Saccharum spontaneum) setts were obtained from Taiwan[66], and planted in soil until flowering. C. sublineola isolate TX430BB was propagated on 5% (w/v) oatmeal agar for 14 days prior to fungal infection experiments. Etiolated sorghum seedlings were prepared as described previously[67]. Briefly, sorghum seeds were germinated in rolls of wet paper towels in dark for 7 days, and were either sprayed with a 0.2% (w/v) bovine gelatin solution (control) or a conidial suspension (1.0 × 10$^6$ spores ml$^{-1}$) of C. sublineola isolate TX430BB in the same gelatin solution

(treatment). Both control and treatment groups were incubated under constant light at room temperature and 100% humidity.

### Plant metabolite extraction

One hundred milligrams of sorghum mesocotyls from fungal infection experiments or mechanically wounded wild sugarcane stalks were harvested for metabolite analysis. All plant tissues were frozen by liquid nitrogen and ground into fine powder using TissueLyser II (QIAGEN, Germany). Two hundred microliters of 80% (v/v) methanol (with 10 µM Apigenin-$d_5$ as internal standard) were added to the samples. Metabolites were extracted by ultra-sonicating the samples on an ice-water bath for 30 min. Samples were passed through a 0.22 µm PTFE membrane filter (Phenomenex, USA) prior to HPLC-QTOF-HRMS analysis described below.

### HPLC-QTOF-HRMS and HPLC-MRM analyses

To screen for OMT activities, purified stilbene products of enzyme assays were separated by a Synergi C18 column (Synergi 4µ Fusion RP 80 Å, 50 × 2 mm, Phenomenex, USA) under a flow rate of 0.5 ml min⁻¹ over a 6 min linear gradient of 10%–90% B (described below). Product detection was achieved with a quadruple-time-of-flight-high resolution mass spectrometer (QTOF-HRMS) X500R system (AB Sciex, China) operating under the information-dependent acquisition (IDA) mode. Meanwhile, purified phenyl-propanoid products of enzyme assays were separated on the same column connected to a HP1100 series HPLC system (Agilent Technologies, USA) linked to an AP3200-QTRAP mass spectrometer (AB Sciex, China) operating under the multiple-reaction-monitoring (MRM) mode. To quantify enzyme assay products, purified reaction products were separated on a Kinetex C18 column (Kinetex 2.6 µm C18 100 Å, 100 × 2.1 mm, Phenomenex, USA) connected to the same HPLC-AP3200-QTRAP-MS/MS system operating under MRM mode. A linear gradient of 10%–90% B under a flow rate of 0.5 ml min⁻¹ over 5 min was used for stilbene analysis; a linear gradient of 2%–75% B under a flow rate of 0.5 ml min⁻¹ over 3 min was used for ferulic acid analysis; a linear gradient of 1%–80% B under a flow rate of 0.3 ml min⁻¹ over 3 min was used for sinapic acid analysis. Total metabolites from all plant tissues were separated by the same Kinetex C18 column connected to the X500R QTOF-HRMS system. A linear gradient under a flow rate of 0.2 ml min⁻¹ over a 20 min linear gradient of 10%–90% B was used for separating sorghum and wild sugarcane metabolites, while a 10-min linear gradient of 10%–90% B under a flow rate of 0.3 ml min⁻¹ was used for separating tobacco metabolites.

To screen for SsSTS activities, purified products of enzyme assays were separated by the same Synergi C18 column from above, under a flow rate of 0.5 ml min⁻¹ over a 5 min linear gradient of 10%–90% B. Product detection was achieved with the same QTOF-HRMS X500R system (AB Sciex, China) operating under the information-dependent acquisition (IDA) mode using positive ionization mode.

In all HPLC-MS analyses, the mobile phase consisted of 0.5% (v/v) formic acid/water (A) and 0.5% (v/v) formic acid/methanol (B). Stilbenes were detected using the positive ionization mode while phenolic acids were detected with the negative ionization mode. Compounds were quantified by integration of peak area using the quantification mode of SCIEX OS (for X500R) or Analyst software version 1.5.2 (for AP3200-QTRAP). Identification of compounds and optimization of MRM parameters were achieved by comparing both retention time and MS/MS spectra (Supplementary Fig. 22) with authentic standards (which were included with each independent LC-MS analysis). LC traces, MS/MS spectra and corresponding MS acquisition parameters were reported in the Source Data file. Extraction blank samples were included in between different sample groups. Two-sided Student's *t*-test or Welch *t*-test (under unequal standard deviations) was used for calculation of statistical

significance, with the exact *p*-values and *t*-test used shown in the Source Data file.

### Gene expression analyses

Total RNA was extracted from sorghum mesocotyls collected at a 24-hour interval for up to 96 h post fungal-treatment and wounded wild sugarcane stalks collected at different time points using the TRIzol method (Invitrogen, USA). Reverse transcription and qRT-PCR were performed using PrimeScript RT reagent kit with gDNA eraser (TaKaRa, Japan) and TB Green premix Ex Taq II kit (TaKaRa, Japan), respectively. Semi qRT-PCR was performed with GoTaq DNA polymerase (Promega, USA). Gene-specific primers used for qRT-PCR experiments were listed in Supplementary Table 9. The housekeeping gene *Sorghum bicolor Eukaryotic Initiation Factor 4A-1* (*SbEIF4α*;[68] XM_002451491) and *Saccharum spontaneum Glyceraldehyde 3-Phosphate Dehydrogenase* (*SsGADPH*; Sspon.08G0001560-1A)[69] were used as internal controls for sorghum and wild sugarcane, respectively.

### Cloning of *SbSTS1, SbOMT4, SbSOMT, SbCOMT, SsSTS, and SsCOMT*

The coding sequences (CDS) of *SbSTS1*, *SbOMT4*, *SbSOMT* and *SbCOMT* were amplified from cDNA prepared from fungal-infected mesocotyls of sorghum genotype BTx623 using gene-specific primers (Supplementary Table 9). The CDS of *SsSTS* and *SsCOMT* were cloned from cDNA prepared from wounded wild sugarcane stalk using gene-specific primers (Supplementary Table 9). To generate recombinant proteins, the CDS of *SsSTS* (full length), *SsCOMT* (full length), *SbCOMT* (encoding residues 2–362), and *SbSOMT* (encoding residues 2–377) were cloned into pET-N-His-TEV (Beyotime) via HiFi DNA Assembly (New England BioLabs, USA) or Gibson Assembly (New England BioLabs, USA). Full length CDS of *SbOMT4* was inserted between BamHI and HindIII restriction sites of pET23a(+) vector (Novagen, Germany). The full length CDSs of *SbSTS1* and *SbSOMT* were individually sub-cloned into the binary vector pEAQ-HT by the Gibson Assembly method (New England BioLabs, USA) using gene-specific primers (Supplementary Table 9).

### Recombinant protein production and site-directed mutagenesis

Desired mutations of *SbSOMT* were introduced into its construct using specific primers (Supplementary Table 9). Expression constructs were transformed into Rosetta™ (DE3) competent *Escherichia coli* cells (Novagen, Germany). Protein expression was induced with 0.1 mM Isopropyl-β-thiogalactopyranoside (IPTG) overnight at 16 °C. *E. coli* cells were harvested, resuspended in lysis buffer (20 mM Tris buffer pH 7.9, 300 mM NaCl, 2.5 mM β-mercaptoethanol), and lysed by sonication. Crude proteins were loaded onto a HisTrap HP column (Cytiva, USA) connected to a ÄKTA pure chromatography system (Cytiva, USA). Column was washed with 15 column volumes of lysis buffer with 50 mM imidazole. Proteins were eluted over a gradient of 50–500 mM imidazole. Eluate fractions with the target OMT were pooled, digested with TEV protease, and simultaneously dialyzed against a low salt buffer (20 mM Tris pH 7.9, 50 mM NaCl, 2.5 mM β-mercaptoethanol) overnight at 4 °C. The dialyzed sample was filtered and flowed through HisTrap to remove His-tagged components. The flowthrough was then loaded to HiTrap Q FF columns (Cytiva, USA) and eluted over a gradient of 50–1000 mM NaCl. Polishing was achieved via size-exclusion chromatography using the storage buffer (100 mM HEPES pH 7.9, 100 mM NaCl) as mobile phase. SsSTS was purified under the same conditions except that the pH was adjusted to 7.0 for all buffers. Proteins were concentrated by ultrafiltration and flash frozen for storage at −80 °C.

### Enzyme assays and enzyme kinetics

For initial screening of OMT activities, purified OMT enzymes (10 µg) were incubated in 100 mM HEPES buffer (pH 7.9), 200 µM

*S*-adenosyl-*L*-methionine (SAM) and 100 μM of phenylpropanoid or stilbene substrates (final volume 200 μl) at 30 °C for 2 h. Enzyme kinetics were determined by incubating 1 μg protein with 100 mM HEPES buffer (pH 7.9), 300 μM SAM, and stilbene concentrations ranging from 5 to 200 μM at 30 °C for 5 min. Similar conditions were used to determine enzyme kinetics of SbSOMT towards SAM, except that SAM concentrations ranging from 5 to 500 μM (while stilbene substrates were kept at 1 mM) were included to accommodate its unexpectedly high $K_d$ and $K_m$ values. For the SsSTS enzyme assays, 10 μg of purified SsSTS enzymes were incubated in 100 mM HEPES buffer (pH 7.0), 100 μM of caffeoyl-CoA or feruloyl-CoA, and 300 μM of malonyl-CoA at 30 °C for 1 h. All enzyme assay reactions were quenched by snap-freezing the reaction tubes in liquid nitrogen. Reaction products were extracted twice with ethyl acetate. The organic layers were then pooled, vacuum-dried, and resuspended in 50 μl 80% (v/v) methanol with 10 μM tricin as internal standard for HPLC-MRM quantification. All reactions were done in triplicates. Enzyme kinetics were calculated using the non-linear regression fitting by GraphPad Prism 6 software (GraphPad, USA).

## Isothermal titration calorimetry (ITC)

Isothermal titration calorimetry (ITC) assays were conducted with MicroCal iTC200 (Malvern Panalytical, UK). Concentrated stocks of resveratrol, pinostilbene, pterostilbene, and piceatannol were prepared in 50% (v/v) PEG 400. Both cell sample and titrant were equilibrated to ITC buffer consisting of 100 mM HEPES pH 7.9 with 2.5% (v/v) PEG 400. For ITC assays involving pterostilbene, PEG 400 concentration was increased to 5% (v/v). Concentrations of protein (in cell) and ligand (in syringe) used in each run are listed in Supplementary Table 9. Each run incorporated an initial delay of 60 s prior to the first injection (0.5 μl) and spaced 180 s (150 s for SsCOMT) between the subsequent 19 injections (2.0 μl) at 25 °C. During the runs, the cell was continuously stirred at 750 rpm by flat paddle. Results were analysed based on the 2nd to 20th injections using the Origin (MicroCal Software, USA) and PEAQ-ITC software (Malvern Panalytical, UK).

## Agroinfiltration of *N. benthamiana* leaves

pEAQ constructs harboring either *SbSTS1* or *SbSOMT* were transformed into *Agrobacterium tumefaciens* strain GV3101. To co-express *SbSTS1* and *SbSOMT* in *N. benthamiana* leaves, two *Agrobacterium* cultures, each harboring one of the overexpression plasmids (individual $OD_{600}$ at 0.8), were mixed in equal ratio and co-infiltrated to *N. benthamiana* leaves. Leaves were harvested five days after infiltration and were subjected to metabolite extraction and HPLC-QTOF-HRMS analysis as described above.

## Constructs of CRISPR-*SbSOMT* and sorghum transformation

The CRISPR-SbSOMT construct was chemically synthesized (Gene Universal, USA). In brief, two gRNAs, including gRNA1: GTTGAA CACGGTGTTCCACG and gRNA2: GCACCGGACTACGCTGTGCG, were designed using CHOPCHOP v3[70], individually introduced to the downstream of *SbU6* promoters, *SbU62.3 and SbU63.1*, respectively, and were further cloned into the plasmid that harbors a selective marker (*NPTII*) to generate CRISPR-SbSOMT construct[50]. The CRISPR/Cas9 plasmid, pBUN411, was modified for sorghum particle bombardment[71]. Sorghum tissue culture, transformation, and CRISPR-Cas9-mediated genome editing were performed as described previously[49]. Briefly, sorghum inbred line Tx430 were grown in a temperature-controlled (18–28 °C) glasshouse to provide the initial explant for transformation. Immature seeds were harvested 11–15 days post anthesis. Immature embryos were isolated and placed on callus induction medium for generating embryogenic calli. The CRISPR-SbSOMT construct and pBUN411 plasmid were co-transformed into embryogenic calli by particle bombardment[72]. After transformation, potential transgenic plantlets were grown in a temperature-controlled

(18–28 °C) glasshouse. Genome-edited plants ($T_0$ and $T_1$) were identified by PCR and direct sequencing using primers listed in Supplementary Table 9.

## X-ray crystallography

SbSOMT was prepared at 3.5 mg ml$^{-1}$ and pre-equilibrated with 0.5 mM of stilbene ligand in a final buffer of 20 mM HEPES pH 7.9, 150 mM NaCl, 4.5 % dimethyl sulfoxide (DMSO) and 1 mM TCEP. Co-crystallization of SbSOMT and resveratrol were initially screened against commercial crystallization screens (Hampton Research, USA and QIAGEN, USA) with a sitting drop vapor diffusion approach incubated at 18 °C. Two crystallizing conditions were further optimized. Crystals of SbSOMT-resveratrol-β-NAD ternary complex were obtained after two days of growing from the setup of 1 μl of 5 mg ml$^{-1}$ SbSOMT, 0.15 μl of 50 mM β-NAD and 0.5 μl of reservoir solution (0.1 M sodium acetate pH 4.6, 0.2 M sodium acetate, 0.2 M NH$_4$Cl, 2.5% (w/v) polyethylene glycol 4000). SbSOMT co-crystalized with pinostilbene, piceatannol, or pterostilbene were obtained under the same condition. Crystal of SbSOMT-resveratrol binary complex was initially obtained after 2–3 weeks of growing from the setup of 1 μl of 5 mg ml$^{-1}$ SbSOMT and 0.5 μl of reservoir solution (0.1 M MES pH 6.5, 0.2 M ammonium sulphate, 30% (w/v) polyethylene glycol monomethyl ether 5000). Subsequent crystallization was accelerated by seeding and crystals were harvested after 1 week. Crystals collected for X-ray diffraction was cryoprotected by 20% (v/v) glycerol topped onto the reservoir solution.

X-ray diffraction data were collected from BL19U1 beamline (wavelength at 0.979 Å) at the Shanghai Synchrotron Radiation Facility and processed by either XDS or HKL3000 package[73,74]. Data were analysed using pipelines provided in CCP4 suite[75]. Data reduction was performed using AIMLESS[76]. The initial structure of SbSOMT-resveratrol-β-NAD ternary complex was solved by molecular replacement; first through PHASER using a chimeric search model generated by MrBUMP, followed by model rebuilding from phase solution through BUCCANEER[77–80]. Following structures were solved by molecular replacement with PHASER using protomer of SbSOMT-resveratrol-β-NAD ternary complex as a search model. The built models were iteratively refined through automated REFMAC5 refinement and manual refinement in COOT[81,82]. Validation and assessment of structure quality were performed using MolProbity and OneDep prior to finalization[83,84]. The statistics and metrics of reported structure are compiled in Supplementary Table 11. Figures featuring protein structure were prepared using UCSF Chimera[85] or UCSF ChimeraX[86].

## Ligand docking with AutoDock Vina

Ligand docking was performed using the default setting[87]. The determination of grid center was guided by superimposition of SbSOMT-resveratrol-β-NAD ternary complex, which the grid center was set in the binding pocket where the superimposed resveratrol is observed. For piceatannol docking in SbSOMT, the grid size was set to 18 Å × 16 Å × 12 Å, covering the entity of superimposed resveratrol. For docking in SbCOMT, a grid size of 24 Å × 24 Å × 24 Å was used with the similar grid center to cover the entire binding pocket. All docking were performed using global searching exhaustiveness of 8 and top 5 docked conformations were recorded for further analysis.

## Phylogenetic analysis

Multiple sequence alignment was done by ClustalW and Clustal Omega with default configurations[88,89]. The unrooted phylogenetic trees were constructed by maximum likelihood method with 1000 bootstrap replicates using MEGA X[90].

## In vitro fungitoxicity assays

A conidial suspension of *Collectotrichum sublineola* isolate TX430BB at a concentration of $5 \times 10^4$ spore ml$^{-1}$ was prepared in double deionized water. Stilbenes including resveratrol, pinostilbene, and pterostilbene

were dissolved in DMSO and supplemented to the potato dextrose agar at concentrations ranging from 0 (mock) to 50 µM. Ten microliters of conidial suspensions were transferred onto the water agar plates, and 5 biological replicates were conducted for each concentration of stilbenes. These plates were sealed, incubated overnight at room temperature in darkness, followed by incubation under constant light for 3 days. Spore germination was observed under a Leica DM500 light microscope (Leica, USA) after incubation in dark. Mycelial growth, represented by the fungal colony diameter, was assessed after the 3-day incubation period under light using the same microscope and the ImageJ software[91]. Data were processed with GraphPad Prism 6 software (GraphPad, USA).

### Accession numbers
Protein sequences analyzed in this study and used in phylogenetic analysis could be found under accession numbers listed in Supplementary Table 12[65].

### Reporting summary
Further information on research design is available in the Nature Portfolio Reporting Summary linked to this article.

## Data availability
Unprocessed metabolite data (including LC traces and MS/MS spectra) of sorghum and wild sugarcane are provided in the Source Data file. Raw LC-MS data are available from the corresponding author (clive-lo@hku.hk) upon request. Atomic coordinates and structure factors for the crystal structures reported in this work were deposited to the Protein Data Bank under accession numbers: 7VB8, 7WAQ, 7WAR and 7WAS. Source data are provided with this paper.

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

## Acknowledgements

We thank the Shanghai Synchrotron Radiation Facility and its staff members for the support. We also thank Prof. George Lomonossoff at John Innes Centre and Leaf Systems International Ltd, for supplying the pEAQ-HT vector. This work was supported by funding from Research Grants Council, University Grants Committee of Hong Kong (grant nos. GRF17110021 and GRF17104720, C.L.; GRF17104120, Q.H.). A.C.W.L. and K.C.P acknowledge the Hong Kong PhD Fellowship Scheme. Publication made possible in part by support from the HKU Libraries Open Access Author Fund sponsored by the HKU Libraries.

## Author contributions

A.C.W.L., K.C.P., Q.H., and C.L. designed experiments. A.C.W.L., K.C.P., N.L., G.L., Z.F., and C.J.K. conducted experiments. A.C. W.L., K.C.P., N.L., L.P.Y.L., Q.H., and C.L. analyzed results. A.C.W.L., K.C.P., L.P.Y.L., Q.H., and C.L. wrote and prepared the manuscript with additional input from I.D.G., Y.T., LW, and other co-authors.

## Competing interests

The authors declare no competing interests.
