## [Peer Review File · Nature Communications]

Regioselective stilbene O-methylations in Saccharinae grassesReviewer #1 (Remarks to the Author):

The manuscript by Lui, et al. describes the identification, biochemical characterization and structural characterization of OMTs (SbSOMT2 and SsCOMT) involved in stilbene biosynthesis. Stilbene compounds were identified in different sorghum cultivars post-infection, with a resistant cultivar producing pinostilbene and pterostilbene (stilbenes methylated at the meta positions). Mechanical wounding of wild sugarcane produced little pinostilbene or pterostilbene and seemed to require a hydroxyl at the para position to perform meta methylation for the production of isorhapontigenin from piceatannol. Based on transcriptomic analysis the OMT involved in stilbene biosynthesis in sorghum was identified (SbSOMT2). CRISPR-Cas9 ko were used to show that SbSOMT2 is required in vivo for the biosynthesis of pinostilbene and pterostilbene in sorghum. The substrate and regioselectivities of the two enzymes (SbSOMT2 and SsCOMT) was demonstrated and crystallographic and modelling performed to understand the different binding modes of the substrates in the binding pocket of the enzymes. Based on these data, SbSOMT2 is a specialized OMT that efficiently produces pino and pterostilbene, whereas the SsCOMT, more closely related to SbCOMT, is not able to efficiently perform the same reactions.

The manuscript presents many different experiments and the characterization of SbSOMT2 and its identification as the enzyme critical for pino and pterostilbene in Sorghum is very solid. The work will be of interest to a specialized community.

Main issues

The abstract is rather confusing as it compares the SbSOMT2 structure (this manuscript) with the SbCOMT structure (previous work by another group) and then discusses SsCOMT (described in this manuscript) and calls it a canonical COMT but does not use the same style acronym (Genus species designation, i.,e. SsCOMT). The main messages seem to be investigations of sorghum and wild sugarcane OMTs in stilbene biosynthesis, but this is a bit difficult to understand from the abstract without a detailed reading of the manuscript and then a re-reading of the abstract.

The use of SbCOMT for comparison and as a proxy for SsCOMT is a bit strange. The enzymatic profiles of SbSOMT2 and SsCOMT could be compared and structural modelling of the SsCOMT done with the SbCOMT structure. It can be difficult to follow the logic of some of the experiments, particularly the use of SbCOMT for enzymatic assays when the "take-home" message of the manuscript seems to be that SbSOMT2 is a specialized OMT for stilbenes that is not present in Saccharum spp. and that recruitment of COMTs by Sorghum spp. occurred after the Sorghum/Saccharum divergence.

Enzyme kinetics

A table of kinetic parameters for SbSOMT2, SbCOMT and SsCOMT should be presented in the main text (Km, kcat, Ki). ITC data should also be presented in a table for clarity. SsCOMT ITC data should be obtained for comparison.

Mass spectrometry

For the mass spec data, what internal standards were used and how many replicates were done (biological and technical)?

SD should be shown. Some of this information is in supplemental, but it would be much more helpful as a simple table presented in the main text.

Fragmentation patterns can be given in supplemental as per Fig. S1.

The Figure 1 panels and fonts are so small they are difficult to read and a summary table would be appreciated. What does Fig 1F show? It is very hard to see anything with respect to phenotypes except the red for SC748-6 96h pi. Are there other phenotypes that should be noted? Anything for BTx623?

Structural studies

Are the structures deposited in the PDB? Please provide validation reports for the structures from this work.

Figures

Fig1 Fonts are very small and the resolution is a bit low. It would be helpful to highlight the methyl group added for the different compounds.

Fig2 This information would be easier to understand in a table format in the main text.

Fig 3b what are the phenotypes? The panel is so small that it is difficult to see anything. The phenotypes should be described in the figure legend.

Fig 4c the pink and blue are very "washed out" and the structural overlays are not particularly informative. Why is the threonine loop depicted as a surface? As this is part of the dimerization interface, it is impossible to see how this contributes with these colors.

Fig4d Again, the colors are so light that the substrate is difficult to see. I cannot identify the conserved leucine and methionine in the wide view, these should be clearly labelled. What are the black dotted lines? This is not described in the figure legend.

Fig5a The mesh is a void or a surface mesh of something? It is very difficult to understand what one is looking at. The binding pocket could be better shown as the mesh seems to simply cover everything.

Fig5b-d please show distances for H-bonds

There is a lot of information in the supplemental section. Some can be moved to the main text in table format. If possible, the most important data should be in the main text and not require going back and forth to supplemental.

Reviewer #2 (Remarks to the Author):

In this manuscript, the authors presented a novel SbSOMT2, which is indispensable for pathogen-inducible 3,5-bis-O-methylated biosynthesis in sorghum and a canonical COMT involved in isorhapontigenin formation in wounded wild sugarcane. At the same time, the authors reported the first crystal structure of a bona fide stilbene O-methyltransferase and revealed the regioselectivities of SbSOMT2 and COMTs. Besides, phylogenetic analysis was applied to explain the divergence of Sorghum spp. from Saccharum spp. The experimental design of this paper is complete and the research content is interesting, however, there are some minor points that should be improved. Here are questions or comments from the reviewer:

1. What is the correlation between the functional study of OMTs in sugarcane and this study, and what is the significance of the work?
2. We notice that the authors examined the metabolic profiles of wild-type sorghum upon infection of Colletotrichum sublineola, whereas, mechanical wounding was selected for metabolic profiling of sugarcane. So why not employ the same treatment for the two plants, and what is the rationale for your choice?
3. Line 43-45: The whole article does not explain why the two species sorghum and sugarcane were chosen. And the evolutionary relationship is not fully elucidated. We would like to know whether two types of OMT can be found simultaneously in the same species.
4. Some content in the text does not match the picture mentioned, eg, line 175-177, "Overexpression of both SbSTS1 and SbSOMT2 produced pterostilbene as the only stilbene product in agro-infiltrated leaves." should correspond to "Supplementary Fig. S5b-d", instead of "Supplementary Fig. S5a-b". Please carefully examine the full text.
5. Line 183: The format of this paragraph is advised to modify, and other parts of the text should also be in a uniform format.

6. Line 197-199: The 72 h and 96 h after infection mentioned in the article are not shown in fig. 3.
7. Line 221-223: Because no suitable OMT can be found, it should not be assumed that SsCOMT is a gene involved in isorhapontigenin biosynthesis, and the results of the correlation test showed that there was no correlation between the two genes. Please explain this part.
8. In Figure 2a-b, it shows that SbSOMT2 and SbCOMT did not obtain the maximum conversion rate and whether it is necessary to continue to prolong the reaction time.
9. In Figure 5, the figure legends of Fig. 5e and Fig. 5f is repeated.

Reviewer #3 (Remarks to the Author):

The manuscript deals with the enzymes that catalyse stilbene O-methylations in Saccharinae grasses. The manuscript describes the biochemical characterisation of several OMTs of two organisms, both with in vitro and in vivo experiments, while they also resolved some crystal structures to understand the catalytic mechanism and the important amino acids for the stilbene binding. Although the manuscript presents many experimental results and it could have a significant impact on the field of biocatalysis, especially in stilbene biosynthesis, in my opinion, it lacks focus, and sufficient depth in each analysis. As I will describe in the detailed comments, the evolutionary discussion is a theory, without any experimental results to support it, while there are some results in the characterisation of the enzymes that do not fit (for instance ITC and docking experiments). Thus, I fail to see the added-value of the characterisation of enzymes of more than one organism, while the discussion of the results of the one organism is not complete. Thus, although I see that there is potential in this work to be published in Nature Communications, I do not think that - in the current form - it can be considered for publication. I would suggest that it requires extensive re-structuring and some more experiments to meet the standards of the journal.

Detailed comments (in order of appearance):

- 1) Lines 156-164: The authors focus on three enzymes, SbSOMT1, SbSOMT2 and SbCOMT. Despite the annotation, SbSOMT1 does not seem to be active on methylation of stilbenes. The authors do not comment on that and the physiological role of this enzyme. Is it a false annotation or does it indeed have a role on stilbene decorations? Are there some other stilbenes that could be converted from this enzyme? What is the substrate scope of homologous enzymes? Moreover, the COMT accepts stilbenes, is it normal, or is it false annotation again? Or maybe a promiscuous activity of this enzyme? The authors do not make any in-depth analysis and discussion on these topics, they just move to the next experiments, and as a reader, I do not get the reason why should I know all these details if they do not lead anywhere. The authors could just say that they identified these three genes of putative OMTs in this organisms and they characterized them and continued their work with the two of them.
- 2) Lines 177-178: Which enzyme is responsible for the "small amount of pinostilbene", do the authors know that *N. benthamiana* has any OMTs that could - via catalytic promiscuity - accept resveratrol as substrate?
- 3) Lines 183-193: I do not get the experimental design. As the mutations are expected to give an alternative reading frame, why did they focus in both exons? If the genetic modification is successful in exon 1, then the exon 2 should not be expressed - at least not with the proper amino acid sequence.
- 4) Lines 195-201: As SbCOMT is not targeted in these variants, and it can methylate resveratrol (from the in vitro experiments), why isn't pinostilbene observed? Did the authors check if the SbCOMT was upregulated after infection?
- 5) Lines 221-223: Any logical expansion why SsCOMT is downregulated, still being involved in the isorhapontigenin biosynthesis? The authors form a hypothesis without any supporting results. If they think that the downregulation of the SsCOMT is irrelevant, why do they comment on that? There should be a reason for the downregulation, they should focus more on that and give convincing hypothesis.

6) Lines 225-234: The authors should provide some structural reasons why there is this difference in substrate scope between SbCOMT and SsCOMT. Which are the enzymes responsible for the strict selectivity for substrate with 4'-OH?

7) Line 243: What do the authors mean by "resemblance"? Sequence similarity or 3D Folding? In what extent?

8) Lines 245-252: The ability of SbCOMT to methylate hydroxyl groups in ring B, suggest that it can accommodate piceatannol in the active site with an inverted orientation. Don't they see any products with methylation in the ring A? The authors should deepen their analysis here. From the respective paragraphs (see also comment 11), I did not understand that their experiments explained the underlying reason behind this opposite docking of the substrate.

9) Lines 254-260: Here, I do not understand the way the authors read their results. Do they state that in some cases, higher K_d values can lead to faster dissociation of the product? first of all, they could perform the ITC experiments with the products too, if they wanted to discuss the dissociation of the product. If this is their theory, they should discuss all the results in the same way, and not only the one that fit with the experimental results. Moreover, if they are interested in the $K_{on} - K_{off}$, maybe they should try another technique, like Surface Plasmon Resonance. I do not get any added-information from this paragraph, it just confuses the reader more.

10) Line 273: Why do the authors crystallize the SbSOMT2 with nicotinamide in a complex? As far as I know NAD is not needed for the catalytic mechanism.

11) Lines 311-315: As a follow up to comment 8, here are the docking experiments of SbCOMT with resveratrol, which, according to the authors, can be docked with two opposite orientations, with similar free energies. This should give methylated products in ring A and ring B, but they mention that the b-ring orientation is not productive? Why is that? (also in lines 421-423 it states that the A-ring methylation is favorable, in the end I am confused...).

12) Lines 334-335: What do the authors compare to? To COMT or to SOMT2?

13) Lines 358-367: The DM mutant is the I144N/F337N of SbSOMT2? It should be clarified in line 359 (as another DM is mentioned in line 334). Also, as mentioned before, the K_d of the products should be checked too, these data will help the discussion of these paragraphs.

14) Lines 369-381: This paragraph has no results. Maybe it is a hypothesis for discussion, but there are no experimental evidences.

Minor comments in Materials and Methods section:

15) Lines 551-554: This belongs to the cloning paragraph.

16) Line 575: What does it mean "(up to 500 μ M for SAM)"? Isn't SAM fixed at 300 μ M?

17) Lines 602-618: They could potential be in one paragraph.

Reviewer #4 (Remarks to the Author):

In this work, Lui et al. have investigated functional and evolutionary aspects of the biosynthesis of stress-related methylated stilbene derivatives in sorghum (*Sorghum bicolor*) and sugarcane (*Saccharum spontaneum*). To this end, the Authors have used an extremely wide array of complementary approaches for the identification and the functional characterization of stilbene O-methyltransferase (SOMT) genes. Indeed, these approaches include genomic and transcriptomic analyses for the identification of candidate genes, detailed characterization of the corresponding recombinant proteins, analysis of in planta activity following expression in *Nicotiana benthamiana*. In addition, a CRISPR/Cas9-mediated genome editing strategy was used for generating the corresponding mutant in sorghum and validate de role of the SbSOMT gene in the biosynthesis of

pterostilbene and in vitro growth inhibition tests were used to evaluate the impact of different stilbenes on the development of the pathogenic fungi *Colletotrichum sublineola*. Finally, the Authors have solved high-resolution crystal structures of SbSOMT2 in order to investigate the molecular features that allowed the evolution of SOMT activity from a caffeic acid O-methyltransferase (COMT) ancestor.

From an evolutionary point of view, this work presents the first example of the direct evolution of SBOMT genes from a COMT ancestor. By contrast, the few SOMT characterized previously from other plant species such as grapevine (VvROMT) and Scots pine (PsPMT2) evolved convergently from non-COMT ancestors. This result is very interesting and unexpected, as sorghum evolved in parallel the SbOMT3 gene, which is more similar to the grapevine and Scots pine SOMTs. However, this root-specific gene is not involved in the methylation of stilbenes, but plays a key role in the biosynthesis of the allelochemicals sorgoleones.

The CRISPR/Cas9 sorghum mutant affected in the biosynthesis of pterostilbene will offer an unprecedented opportunity to decipher the biological and ecological roles of methylated stilbenes in the resistance against pathogens. This aspect will undoubtedly be the subject of future work.

As a summary, this study presents an elegant, thorough and well-conducted combination of functional genomic approaches that greatly adds to our knowledge the evolution of the biosynthesis of O-methylated stilbenes.

Minor comment

Sup. Fig. 5: misspelling of resveratrol in panel b.

REVIEWER COMMENTS

Reviewer #1 (Remarks to the Author):

The manuscript by Lui, et al. describes the identification, biochemical characterization and structural characterization of OMTs (SbSOMT2 and SsCOMT) involved in stilbene biosynthesis. Stilbene compounds were identified in different sorghum cultivars post-infection, with a resistant cultivar producing pinostilbene and pterostilbene (stilbenes methylated at the meta positions). Mechanical wounding of wild sugarcane produced little pinostilbene or pterostilbene and seemed to require a hydroxyl at the para position to perform meta methylation for the production of isorhapontigenin from piceatannol. Based on transcriptomic analysis the OMT involved in stilbene biosynthesis in sorghum was identified (SbSOMT2). CRISPR-Cas9 ko were used to show that SbSOMT2 is required in vivo for the biosynthesis of pinostilbene and pterostilbene in sorghum. The substrate and regioselectivities of the two enzymes (SbSOMT2 and SsCOMT) was demonstrated and crystallographic and modelling performed to understand the different binding modes of the substrates in the binding pocket of the enzymes. Based on these data, SbSOMT2 is a specialized OMT that efficiently produces pino and pterostilbene, whereas the SsCOMT, more closely related to SbCOMT, is not able to efficiently perform the same reactions.

The manuscript presents many different experiments and the characterization of SbSOMT2 and its identification as the enzyme critical for pino and pterostilbene in Sorghum is very solid. The work will be of interest to a specialized community.

Main issues

The abstract is rather confusing as it compares the SbSOMT2 structure (this manuscript) with the SbCOMT structure (previous work by another group) and then discusses SsCOMT (described in this manuscript) and calls it a canonical COMT but does not use the same style acronym (Genus species designation, i.e. SsCOMT). The main messages seem to be investigations of sorghum and wild sugarcane OMTs in stilbene biosynthesis, but this is a bit difficult to understand from the abstract without a detailed reading of the manuscript and then a re-reading of the abstract.

The use of SbCOMT for comparison and as a proxy for SsCOMT is a bit strange. The enzymatic profiles of SbSOMT2 and SsCOMT could be compared and structural modelling of the SsCOMT done with the SbCOMT structure. It can be difficult to follow the logic of some of the experiments, particularly the use of SbCOMT for enzymatic assays when the “take-home” message of the manuscript seems to be that SbSOMT2 is a specialized OMT for stilbenes that is not present in Saccharum spp. and that recruitment of COMTs by Sorghum spp. occurred after the Sorghum/Saccharum divergence.

Response:

Thank you for your comment. We have re-organized the abstract and manuscript accordingly to address your concerns, and renamed the proteins accordingly for coherence. Gene/protein nomenclatures have also been formatted using the same style. SbSOMT2 is now renamed SbSOMT while SbOMT1 is renamed SbOMT4 as it does not show SOMT activities.

Since SbSOMT is derived from an ancestral grass COMT and it shows catalytic regioselectivity different from that of SbCOMT, our work primarily focuses on multifaceted characterizations

and direct comparisons between the two sorghum enzymes. Following structural analyses, we uncovered the molecular basis for the opposite substrate binding orientations that account for their respective catalytic regioselectivity. Our further investigations suggested that the piceatannol-derived stilbene profile in wild sugarcane is likely contributed by a canonical COMT, such as SsCOMT which showed the same catalytic regioselectivity as SbCOMT, in the absence of a SOMT ortholog.

Enzyme kinetics

A table of kinetic parameters for SbSOMT2, SbCOMT and SsCOMT should be presented in the main text (K_m , k_{cat} , K_i). ITC data should also be presented in a table for clarity. SsCOMT ITC data should be obtained for comparison.

Response:

Thank you for your comment. Both kinetic and ITC data of SbSOMT and SbCOMT are now presented in the main text (Table 1 & 2), while the ITC data of SsCOMT are included in Supplemental Table 6.

Mass spectrometry

For the mass spec data, what internal standards were used and how many replicates were done (biological and technical)? SD should be shown.

Response:

Thank you for your comment. For the mass spectrometry data of plant metabolites including those from sorghum, tobacco, and sugarcane, an internal standard of Apigenin-*d*5 was included in our extraction buffer and is described at Lines 554-556. Three biological replicates were included for each of these experiments. For the mass spectrometry data of *in vitro* enzyme assays and enzyme kinetics studies, an internal standard of Tricin was included in our extraction buffer and is described at Lines 649-651. Three technical replicates were included for each of these experiments. SD are also shown in graphs and tables. The above information is now included in the legend of relevant items.

Some of this information is in supplemental, but it would be much more helpful as a simple table presented in the main text.

Response:

Thank you for your comment. We have revised the manuscript accordingly and moved/integrated the following key supplemental data to the main text: summaries of stilbene profile at experimental end points for sorghum (now in Fig. 1b) and wild sugarcane (now in Fig. 7a); enzyme kinetics of SbCOMT and SbSOMT towards pinostilbene (now as Table 1), and their ITC data towards different stilbenes (now as Table 2).

Fragmentation patterns can be given in supplemental as per Fig. S1.

Response:

Thank you for your comment. We have now included a summary of MS/MS fragmentation pattern of different stilbene compounds from samples and corresponding authentic standards in Supplemental Fig. S22.

The Figure 1 panels and fonts are so small they are difficult to read and a summary table would

be appreciated. What does Fig 1F show? It is very hard to see anything with respect to phenotypes except the red for SC748-6 96h pi. Are there other phenotypes that should be noted? Anything for BTx623?

Response:

Thank you for your comment. We have enlarged the panels and fonts to improve readability of Figure 1. In addition, Fig.1f has been removed in the current version of manuscript. The phenotypes of SC748-6 and BTx623 have been published previously (Du *et al.*, 2010, Liu *et al.*, 2010).

Structural studies

Are the structures deposited in the PDB? Please provide validation reports for the structures from this work.

Response:

Thank you for your comment. We have uploaded the PDB validation reports for further reference. In addition, all SbSOMT structural data and associated reports are now available at Protein Data Bank under the following IDs: 7WAQ, 7VB8, 7WAR, and 7WAS.

Figures

Fig1 Fonts are very small and the resolution is a bit low. It would be helpful to highlight the methyl group added for the different compounds.

Response:

Thank you for your comment. We have improved the resolution in the revised figure. The added methyl groups are now color-coded for clarity.

Fig2 This information would be easier to understand in a table format in the main text.

Response:

Thank you for your comment. We have now used bar charts to show the abundance of each enzyme assay product at the two-hour end point for clarity (Fig. 2a-c).

Fig 3b what are the phenotypes? The panel is so small that is it difficult to see anything. The phenotypes should be described in the figure legend.

Response:

Thank you for your comment. Fig. 3b has been removed in the current version of manuscript.

Fig 4c the pink and blue are very “washed out” and the structural overlays are not particularly informative. Why is the threonine loop depicted as a surface? As this is part of the dimerization interface, it is impossible to see how this contributes with these colors.

Response:

Thank you for your comment. We have reconfigured all relevant figures for better contrast and clarity. For Fig. 4c (moved to Fig. 4d in the current version), we hope that it is now properly displaying the following key points: highly similar global structures of SbCOMT and SbSOMT,

ligand orientations for SbSOMT, and the unique SbSOMT Thr-rich loop that is absent in SbCOMT.

Fig4d Again, the colors are so light that the substrate is difficult to see. I cannot identify the conserved leucine and methionine in the wide view, these should be clearly labelled. What is the black dotted lines? This is not described in the figure legend.

Response:

Thank you for your comment. Since NAD was only an additive to facilitate protein crystallization and do not contribute to OMT activity, we have decided to remove Fig. 4d together with the corresponding text in the current version of manuscript.

Fig5a The mesh is a void or a surface mesh of something? It is very difficult to understand what one is looking. The binding pocket could be better shown as the mesh seems to simply cover everything.

Response:

Thank you for your comment. We have removed the mesh of the binding pocket for better contrast and clarity.

Fig5b-d please shown distances for H-bonds

Response:

Thank you for your comment. Distances for H-bonds are now displayed in Fig. 5b-d.

There is a lot of information in the supplemental section. Some can be moved to the main text in table format. If possible, the most important data should be in the main text and not require going back and forth to supplemental.

Response:

Thank you for your comment. We have revised the manuscript accordingly and moved/integrated the following key supplemental data to the main text: summaries of stilbene profile at experimental end points for sorghum (now in Fig. 1b) and wild sugarcane (now in Fig. 7a); enzyme kinetics of SbCOMT and SbSOMT towards pinostilbene (now in Table 1), and their ITC data towards different stilbenes (now in Table 2).

References:

- Liu, H. *et al.* Molecular dissection of the pathogen-inducible 3-deoxyanthocyanidin biosynthesis pathway in sorghum. *Plant Cell Physiol* **51**, 1173-1185 (2010).
- Du, Y., Chu, H., Wang, M., Chu, I. K. & Lo, C. Identification of flavone phytoalexins and a pathogen-inducible flavone synthase II gene (SbFNSII) in sorghum. *J Exp Bot* **61**, 983-994 (2010).

Reviewer #2 (Remarks to the Author):

In this manuscript, the authors presented a novel SbSOMT2, which is indispensable for pathogen-inducible 3,5-bis-O-methylated biosynthesis in sorghum and a canonical COMT involved in isorhaphontigenin formation in wounded wild sugarcane. At the same time, the authors reported the first crystal structure of a bona fide stilbene O-methyltransferase and revealed the regioselectivities of SbSOMT2 and COMTs. Besides, phylogenetic analysis was applied to explain the divergence of Sorghum spp. from Saccharum spp. The experimental design of this paper is complete and the research content is interesting, however, there are some minor points that should be improved. Here are questions or comments from the reviewer:

1. What is the correlation between the functional study of OMTs in sugarcane and this study, and what is the significance of the work?

Response:

Thank you for your comment. Since SbSOMT is derived from an ancestral grass COMT and it shows catalytic regioselectivity different from that of SbCOMT, our work primarily focuses on multifaceted characterizations and direct comparisons between the two sorghum enzymes. Following structural analyses, we uncovered the molecular basis for the opposite substrate binding orientations that account for their respective catalytic regioselectivity. Our further investigations suggested that the piceatannol-derived stilbene profile in wild sugarcane is likely contributed by a canonical COMT, such as SsCOMT which showed the same catalytic regioselectivity as SbCOMT, in the absence of a SOMT.

2. We notice that the authors examined the metabolic profiles of wild-type sorghum upon infection of *Colletotrichum sublineola*, whereas, mechanical wounding was selected for metabolic profiling of sugarcane. So why not employ the same treatment for the two plants, and what is the rationale for your choice?

Response:

Thank you for your comment. Our preliminary experiment showed that *C. sublineola* could not infect wild sugarcane tissues (data not shown), including both leaf and stalk. Meanwhile, mechanical wounding has been reported to induce piceatannol accumulation in sugarcane spp., making it an obvious choice to test for stilbene O-methylation in wild sugarcane.

3. Line 43-45: The whole article does not explain why the two species sorghum and sugarcane were chosen. And the evolutionary relationship is not fully elucidated. We would like to know whether two types of OMT can be found simultaneously in the same species.

Response:

Thank you for your comment. We have now included the rationales of choosing sorghum and wild sugarcane (e.g. Lines 102-106, 378-382, 492-496). Briefly, both species are closely related, belonging to the same subtribe (Saccharinae) with highly conserved orthologs such as STS, COMT, etc. Both of them produce stilbenes but they are O-methylated in different rings. Both COMT and SOMT are present simultaneously in sorghum while COMT but not SOMT is present in sugarcane. Our structural analysis has revealed the molecular basis underlying the different regioselectivities for COMT and SOMT, hence rationalizing the different O-methylated stilbene profiles in sorghum and sugarcane.

4. Some content in the text does not match the picture mentioned, eg, line 175-177,

“Overexpression of both SbSTS1 and SbSOMT2 produced pterostilbene as the only stilbene product in agro-infiltrated leaves.” should correspond to “Supplementary Fig. S5b-d”, instead of “Supplementary Fig. S5a-b”. Please carefully examine the full text.

Response:

Thank you for your comment. We have corrected this error (now at Supplementary Fig. S3a-c) and double-checked all references to figures.

5. Line 183: The format of this paragraph is advised to modify, and other parts of the text should also be in a uniform format.

Response:

Thank you for your comment. We have revised and simplified this paragraph accordingly (now at Line 188-193).

6. Line 197-199: The 72 h and 96 h after infection mentioned in the article are not shown in fig. 3.

Response:

Thank you for your comment. The LC-MS chromatograms shown in Fig. 3 were derived from tissues 96 h after infection. This information is now added in the figure legend. In addition, we have summarized details of these data for both time points (72 h and 96 h post-infection) in Supplementary Table 4.

7. Line 221-223: Because no suitable OMT can be found, it should not be assumed that SsCOMT is a gene involved in isorhapontigenin biosynthesis, and the results of the correlation test showed that there was no correlation between the two genes. Please explain this part.

Response:

Thank you for your comment. We agree that further investigations involving genetic evidence are required to establish the role of SsCOMT in isorhapontigenin biosynthesis in wild sugarcane. We have revised the manuscript to explicitly express this point (e.g. Line 492-496) and toned down the wordings throughout the manuscript accordingly. Nonetheless, our catalytic and structural analyses strongly indicated that COMTs accept piceatannol for B-ring *O*-methylation to produce isorhapontigenin. In wild sugarcane, SsCOMT is still constitutively expressed despite its down-regulation upon wounding. When piceatannol becomes increasingly available following wounding, it is likely that small amounts of them are converted to isorhapontigenin by SsCOMT. Actually, isorhapontigene only constituted approximately 6% of stilbene aglycones produced in wounded wild sugarcane.

8. In Figure 2a-b, it shows that SbSOMT2 and SbCOMT did not obtain the maximum conversion rate and whether it is necessary to continue to prolong the reaction time.

Response:

Thank you for your comment. The original Fig. 2a-b showed and compared product formation over time by SbCOMT and SbSOMT when incubated with resveratrol or phenylpropanoid substrates up to 2 hours and were not used for calculation of Michaelis–Menten enzyme kinetic parameters (which were calculated based on a 5-minute incubation period with various pinostilbene concentrations and now shown in Table 1). We have now replaced these two

figures with Fig. **2a-c** which show the abundance of each product at the two-hour end point for clarity.

9. In Figure 5, the figure legends of Fig. 5e and Fig. 5f is repeated.

Response:

Thank you for spotting the error. We have revised the figure legend of Fig. **5e** and **5f** accordingly.

Reviewer #3 (Remarks to the Author):

The manuscript deals with the enzymes that catalyse stilbene O-methylations in Saccharinae grasses. The manuscript describes the biochemical characterisation of several OMTs of two organisms, both with in vitro and in vivo experiments, while they also resolved some crystal structures to understand the catalytic mechanism and the important amino acids for the stilbene binding. Although the manuscript presents many experimental results and it could have a significant impact on the field of biocatalysis, especially in stilbene biosynthesis, in my opinion, it lacks focus, and sufficient depth in each analysis. As I will describe in the detailed comments, the evolutionary discussion is a theory, without any experimental results to support it, while there are some results in the characterisation of the enzymes that do not fit (for instance ITC and docking experiments). Thus, I fail to see the added-value of the characterisation of enzymes of more than one organism, while the discussion of the results of the one organism is not complete. Thus, although I see that there is potential in this work to be published in Nature Communications, I do not think that - in the current form - it can be considered for publication. I would suggest that it requires extensive re-structuring and some more experiments to meet the standards of the journal.

Detailed comments (in order of appearance):

1) Lines 156-164: The authors focus on three enzymes, SbSOMT1, SbSOMT2 and SbCOMT. Despite the annotation, SbSOMT1 does not seem to be active on methylation of stilbenes. The authors do not comment on that and the physiological role of this enzyme. Is it a false annotation or does it indeed has a role on stilbene decorations? Are there some other stilbenes that could be converted from this enzyme? What is the substrate scope of homologous enzymes?

Response:

Thank you for your comment. We concur with your comment regarding SbSOMT1 which is now renamed SbOMT4 as it is only a putative OMT. As kindly suggested by you below, we now mention that three OMT candidates were identified and continued our work with SbSOMT and SbCOMT as they showed activities in our enzyme assays.

Moreover, the COMT accepts stilbenes, is it normal, or is it false annotation again? Or maybe a promiscuous activity of this enzyme? The authors do not make any in-depth analysis and discussion on these topics, they just move to the next experiments, and as a reader, I do not get the reason why should I know all these details if they do not lead anywhere. The authors could just say that they identified these three genes of putative OMTs in this organisms and they characterized them and continued their work with the two of them.

Response:

Thank you for your comments. Canonical COMTs generally show a great extent of substrate promiscuity and different names may be assigned to the same COMT which causes confusion for readers. For instance, rice OsCOMT (also known as OsROMT9 or OsCAldOMT) catalyzes the O-methylations of both flavonoids and hydroxycinnamic acids (including caffeic acid and its derivatives) (Lam *et al.*, 2019). Multifunctional Arabidopsis AtCOMT catalyzes O-methylations of flavonoids (Muzac *et al.*, 2000), hydroxycinnamic acids (Goujon *et al.*, 2003), and N-acetylserotonin (Byeon *et al.*, 2015) and is thus referred to as AtOMT1/AtOMT3/AtASMT in different works. Similarly, while SbCOMT and SsCOMT can utilize stilbenes as substrates, their nomenclature and functions as canonical COMTs

(especially for SbCOMT) have been widely established in literature (e.g. Green *et al.*, 2014, Louie *et al.*, 2010, Saluja *et al.*, 2021). Thus, we have decided to keep their nomenclature to avoid confusion.

2) Lines 177-178: Which enzyme is responsible for the "small amount of pinostilbene", do the authors know that *N. benthamiana* has any OMTs that could - via catalytic promiscuity - accept resveratrol as substrate?

Response:

Thank you for your comment. It is unclear that which *N. benthamiana* OMT(s) could utilize resveratrol as substrate.

3) Lines 183-193: I do not get the experimental design. As the mutations are expected to give an alternative reading frame, why did they focus in both exons? If the genetic modification is successful in exon 1, then the exon 2 should not be expressed - at least not with the proper amino acid sequence.

Response:

Thank you for your comment. It is very common to design multiple sgRNA target sites for the same gene to maximize the possibility of obtaining useful mutants for subsequent analyses, especially for species that are recalcitrant to genetic transformation such as sorghum. In our case, both sgRNA targets were successfully mutated although the one located at exon 1 alone was sufficient in inducing frameshift truncation of SbSOMT in all mutant lines. Nonetheless, we have revised this paragraph (now at Lines **188-193**) to concisely describe this key mutation at exon 1.

4) Lines 195-201: As SbCOMT is not targeted in these variants, and it can methylate resveratrol (from the *in vitro* experiments), why isn't pinostilbene observed? Did the authors check if the SbCOMT was upregulated after infection?

Response:

Thank you for your comment. The *in vitro* experiments provided the "optimal" conditions for the reaction such as the abundant amount of (10 μ g of OMTs) and substrates, and the lack of other competing enzymes (e.g. UDP-glucose transferases) which may compete with SbCOMT to convert resveratrol into piceid. In fact, all three mutant lines accumulated significantly more piceid than the wild-type control 96 h after infection (Supplementary Table 4). Altogether, these sub-optimal *in planta* factors may contribute to a low/zero pinostilbene level in these mutant lines.

5) Lines 221-223: Any logical explanation why SsCOMT is downregulated, still being involved in the isohapontigenin biosynthesis? The authors form a hypothesis without any supporting results. If they think that the downregulation of the SsCOMT is irrelevant, why do they comment on that? There should be a reason for the downregulation, they should focus more on that and give convincing hypothesis.

Response:

Thank you for your comment. First, our catalytic and structural analyses strongly indicated that COMTs accept piceatannol for B-ring *O*-methylation to produce isorhapontigenin. We have re-examined the expression pattern of *SsCOMT* following the wounding treatment by semi-qRT-PCR (now in Supplementary Fig. **S13d**), which showed that *SsCOMT* was constitutively

expressed in the stalk tissues despite its downregulation. When piceatannol becomes increasingly available following wounding, it is likely that small amounts of them are converted to isorhapontogenin by SsCOMT. Actually, isorhapontigenin only constituted approximately 6% of stilbene aglycones produced in wounded wild sugarcane (now shown in Fig. 7a with full details in Supplementary Table 6).

6) Lines 225-234: The authors should provide some structural reasons why there is this difference in substrate scope between SbCOMT and SsCOMT. Which are the enzymes responsible for the strict selectivity for substrate with 4'-OH?

Response:

Thank you for your comment. There are no differences in substrate scope between SbCOMT and SsCOMT *in vitro*. On the other hand, the *in planta* stilbene profiles are different between sorghum and wild sugarcane. Sorghum produces resveratrol but not piceatannol. SbSOMT, but not SbCOMT, catalyzes A-ring *O*-methylations of resveratrol to produce pinostilbene and pterostilbene. Wild sugarcane produces both resveratrol and piceatannol. Since it does not have a SOMT ortholog, no A-ring *O*-methylated stilbenes are produced. Instead, piceatannol is accepted as a substrate for B-ring *O*-methylation by SsCOMT, leading to the accumulation of isorhapontigenin. The above explanations now appear at Lines 492-506.

7) Line 243: What do the authors mean by "resemblance"? Sequence similarity or 3D Folding? In what extent?

Response:

Thank you for your comment. We have revised the wording (now Line 268-269) to "All complexes were in high structural resemblance and depicted as a homodimer with an open conformation." for clarity.

8) Lines 245-252: The ability of SbCOMT to methylate hydroxyl groups in ring B, suggest that it can accommodate piceatannol in the active site with an inverted orientation. Don't they see any products with methylation in the ring A? The authors should deepen their analysis here. From the respective paragraphs (see also comment 11), I did not understand that their experiments explained the underlying reason behind this opposite docking of the substrate.

Response:

Thank you for your comment. This paragraph (now at Lines 240-259) describes the use of piceatannol as substrate for SbSOMT and SbCOMT to demonstrate their differences in catalytic regioselectivity, while in-depth structural and docking analyses (Lines 276-374) further elucidated the underlying molecular mechanism (i.e. differences in key interacting residues like I144N and F337N between SbCOMT and SbSOMT) that accounts for their catalytic regioselectivity. In summary, SbCOMT, through H-bond interactions between its Asn residues and different -OH groups on piceatannol, preferentially orientates the stilbene B-ring towards its active site and produces isorhapontigenin. A very small portion of isorhapontigenin formed was further methylated at the stilbene A-ring to form 3'-methoxypinostilbene but only under *in vitro* conditions (with a low conversion rate that is similar to that of resveratrol-to-pinostilbene reaction catalyzed by SbCOMT as shown in Fig. 2a). No 3'-methoxypinostilbene was detected from wounded stalks of wild sugarcane. Furthermore, SbCOMT could not produce 3'-hydroxypinostilbene or 3'-hydroxypterostilbene from piceatannol through sequential stilbene A-ring *O*-methylations like SbSOMT. In contrast, SbSOMT harbors

hydrophobic residues like Ile144 and Phe337 which greatly favors binding of piceatannol with the opposite orientation (A-ring) to produce 3'-hydroxypinostilbene or 3'-hydroxypterostilbene through sequential stilbene A-ring *O*-methylations as shown in the *in vitro* assays (Fig. 3c) and docking with piceatannol (Fig. 5f).

9) Lines 254-260: Here, I do not understand the way the authors read their results. Do they state that in some cases, higher K_d values can lead to faster dissociation of the product? first of all, they could perform the ITC experiments with the products too, if they wanted to discuss the dissociation of the product. If this is their theory, they should discuss all the results in the same way, and not only the one that fit with the experimental results. Moreover, if they are interested in the K_{on} - K_{off} , maybe they should try another technique, like Surface Plasmon Resonance. I do not get any added-information from this paragraph, it just confuses the reader more.

Response:

Thank you for your comment. Pterostilbene, as the final product between SbSOMT-resveratrol reaction, demonstrates a significantly higher K_d value amongst resveratrol, pinostilbene, and pterostilbene (which are stilbene metabolites found in fungal-infected sorghum mesocotyls) for SbSOMT. This indicates that SbSOMT favors binding of substrates (resveratrol and pinostilbene) over the final product (pterostilbene). Meanwhile, SbCOMT showed no significant difference in K_d towards the same substrates. These comparisons were made in the original manuscript but has been revised to improve clarity (now at Lines 223-238). This section also indicates that as SbSOMT and SbCOMT both displayed strong, micromolar binding affinities towards resveratrol and pinostilbene, which would not account for their differences in catalytic regioselectivity.

10) Line 273: Why do the authors crystallize the SbSOMT2 with nicotinamide in a complex? As far as I know NAD is not needed for the catalytic mechanism.

Response:

Thank you for your comment. It is common to use an additive (such as NAD) during the optimization of crystallization to obtain better crystal and consequently producing better X-ray diffraction data and structure (Cudney *et al.*, 1994). Indeed, NAD is not involved in the catalytic mechanism. We now describe the inclusion of NAD as an additive in protein-substrate crystals for clarity (now at Lines 263-268: We solved the crystal structure... **β -NAD was utilized as an additive to improve crystallization⁴²**).

11) Lines 311-315: As a follow up to comment 8, here are the docking experiments of SbCOMT with resveratrol, which, according to the authors, can be docked with two opposite orientations, with similar free energies. This should give methylated products in ring A and ring B, but they mention that the b-ring orientation is not productive? Why is that?

Response:

Thank you for your comment. Unlike the piceatannol B-ring, the resveratrol B-ring does not contain the 3'-OH group for *O*-methylation by SbCOMT (Fig. 5e-f). It has also been widely reported that canonical COMTs from various species could not methylate the *para*-OH group (i.e. 4'-OH group in the stilbene B-ring). Instead, this 4'-OH group serves to interact with the key residue in COMT to maintain the substrate orientation for 3' -*O*-methylation (Green *et al.*, 2014, Louie *et al.*, 2010, Rimando *et al.*, 2012). In the absence of the 3'-OH group, the B-ring orientation is not catalytically productive for the SbCOMT-resveratrol combination.

(also in lines 421-423 it states that the A-ring methylation is favorable, in the end I am confused...).

Response:

Response: Thank you for your comment and we apologize for the confusion. Our study demonstrated that the favourable reaction, i.e. stilbene A-ring *O*-methylation, by SbSOMT could be attributed to the presence of hydrophobic residues such as Ile144 and Phe337. On the other hand, SbCOMT harbors hydrophilic Asn residues at the equivalent positions which favor B-ring methylation as mentioned above.

The original sentence at Lines **421-423** discussed that the recruitment of hydrophobic residues (equivalent to Ile144 and Phe337 in SbSOMT) likely represented a key molecular event during the acquisition of SOMT activities by an ancestral COMT protein, thus facilitating an opposite orientation for stilbene A-ring *O*-methylation. To avoid confusion, this sentence has been revised accordingly (now at Lines **455-459**).

12) Lines 334-335: What do the authors compare to? To COMT or to SOMT2?

Response:

Thank you for your comment. Comparisons were made between wild-type SbSOMT protein to its corresponding mutants to demonstrate the importance of His282 in SbSOMT substrate binding.

13) Lines 358-367: The DM mutant is the I144N/F337N of SbSOMT2? It should be clarified in line 359 (as another DM is mentioned in line 334). Also, as mentioned before, the K_d of the products should be checked too, these data will help the discussion of these paragraphs.

Response:

Thank you for your comment. We have removed the term “DM” throughout the manuscript for clarity. All mutant proteins are now named according to their mutations, e.g. SbSOMT I144N/F337N mutant protein. The K_d of all SbSOMT mutant proteins are now summarized in Supplementary Table 5.

14) Lines 369-381: This paragraph has no results. Maybe it is a hypothesis for discussion, but there are no experimental evidences.

Response:

Thank you for your comment. This paragraph (now at Lines **205-220**) primarily describes the results from the phylogenetic analysis of SbSOMT with COMTs from diverse species and selected OMTs (including grapevine SOMT and pine SOMT). We believe it is important to understand the phylogenetic relationships between SbSOMT and related enzymes in order to rationalize our subsequent experimental analyses. For example, as SbSOMT is derived from an ancestral grass COMT but shows different catalytic regioselectivity, we focused on multifaceted characterizations and direct comparisons between the two sorghum enzymes. Following structural analyses, we uncovered the molecular basis for the opposite substrate binding orientations that account for their respective catalytic regioselectivity. Our further investigations suggested that the piceatannol-derived stilbene profile in wild sugarcane is likely contributed by a canonical COMT, such as SsCOMT which showed the same catalytic regioselectivity as SbCOMT, in the absence of a SOMT ortholog.

Minor comments in Materials and Methods section:

15) Lines 551-554: This belongs to the cloning paragraph.

Response:

Thank you for your comment. This section (now at Lines **605-610**) has been moved to the cloning paragraph.

16) Line 575: What does it mean "(up to 500µM for SAM)"? Isn't SAM fixed at 300 µM?

Response:

Thank you for your comment. To determine enzyme kinetics towards stilbenes, SAM was fixed at 300 µM while stilbene concentrations ranging from 5 to 200 µM were used. However, as both K_m and K_d values of SbSOMT towards SAM were unexpectedly high, we had to increase SAM (as the target substrate) concentration to up to 500 µM to accurately determine its kinetic parameters. This information can now be found at Lines **642-645** in the current manuscript.

17) Lines 602-618: They could potential be in one paragraph.

Response:

Thank you for your comment. They have been combined into one paragraph (now at Line **685-694**).

References:

- Byeon, Y., Choi, G. H., Lee, H. Y. & Back, K. Melatonin biosynthesis requires *N*-acetylserotonin methyltransferase activity of caffeic acid *O*-methyltransferase in rice. *J. Exp. Bot.* **66**, 6917-6925 (2015).
- Cudney, R., Patel, S., Weisgraber, K., Newhouse, Y., & McPherson, A. Screening and optimization strategies for macromolecular crystal growth. *Acta Crystallogr. D Biol. Crystallogr.* **50**, 414-423 (1994).
- Goujon, T. *et al.* A new *Arabidopsis thaliana* mutant deficient in the expression of *O*-methyltransferase impacts lignins and sinapoyl esters. *Plant Mol. Biol.* **51**, 973-989 (2003).
- Green, A. R. *et al.* Determination of the structure and catalytic mechanism of *Sorghum bicolor* caffeic acid *O*-methyltransferase and the structural impact of three *brown midrib12* mutations. *Plant Physiol.* **165**, 1440-1456 (2014).
- Muzac, I., Wang, J., Anzellotti, D., Zhang, H. & Ibrahim, R. K. Functional expression of an *Arabidopsis* cDNA clone encoding a flavonol 3'-*O*-methyltransferase and characterization of the gene product. *Archives of Biochemistry and Biophysics*, **375**, 385-388 (2000).
- Lam P. Y. *et al.* OsCALdOMT1 is a bifunctional *O*-methyltransferase involved in the biosynthesis of triclin-lignins in rice cell walls. *Sci. Rep.* **9**, 11597 (2019)..
- Louie, G. V. *et al.* Structure-function analyses of a caffeic acid *O*-methyltransferase from perennial ryegrass reveal the molecular basis for substrate preference. *Plant Cell* **22**, 4114-4127 (2010).
- Saluja, M., Zhu, F., Yu, H., Walia, H. & Sattler, S. E. Loss of COMT activity reduces lateral root formation and alters the response to water limitation in sorghum *brown midrib (bmr) 12* mutant. *New Phytol.* **229**, 2780-2794 (2021).

Reviewer #4 (Remarks to the Author):

In this work, Lui et al. have investigated functional and evolutionary aspects of the biosynthesis of stress-related methylated stilbene derivatives in sorghum (*Sorghum bicolor*) and sugarcane (*Saccharum spontaneum*). To this end, the Authors have used an extremely wide array of complementary approaches for the identification and the functional characterization of stilbene O-methyltransferase (SOMT) genes. Indeed, these approaches include genomic and transcriptomic analyses for the identification of candidate genes, detailed characterization of the corresponding recombinant proteins, analysis of in planta activity following expression in *Nicotiana benthamiana*. In addition, a CRISPR/Cas9-mediated genome editing strategy was used for generating the corresponding mutant in sorghum and validate de role of the SbSOMT gene in the biosynthesis of pterostilbene and in vitro growth inhibition tests were used to evaluate the impact of different stilbenes on the development of the pathogenic fungi *Colletotrichum sublineola*. Finally, the Authors have solved high-resolution crystal structures of SbSOMT2 in order to investigate the molecular features that allowed the evolution of SOMT activity from a caffeic acid O-methyltransferase (COMT) ancestor.

From an evolutionary point of view, this work presents the first example of the direct evolution of SBOMT genes from a COMT ancestor. By contrast, the few SOMT characterized previously from other plant species such as grapevine (VvROMT) and Scots pine (PsPMT2) evolved convergently from non-COMT ancestors. This result is very interesting and unexpected, as sorghum evolved in parallel the SbOMT3 gene, which is more similar to the grapevine and Scots pine SOMTs. However, this root-specific gene is not involved in the methylation of stilbenes, but plays a key role in the biosynthesis of the allelochemicals sorgoleones.

The CRISPR/Cas9 sorghum mutant affected in the biosynthesis of pterostilbene will offer an unprecedented opportunity to decipher the biological and ecological roles of methylated stilbenes in the resistance against pathogens. This aspect will undoubtedly be the subject of future work.

As a summary, this study presents an elegant, thorough and well-conducted combination of functional genomic approaches that greatly adds to our knowledge the evolution of the biosynthesis of O-methylated stilbenes.

Responses:

We greatly appreciate your kind and insightful comments on this manuscript.

Minor comment:

Sup. Fig. 5: misspelling of resveratrol in panel b.

Response:

Thank you for your comment. An updated version of this figure can now be found at Supp. Fig. **S3b**.

Reviewer #1 (Remarks to the Author):

The authors have addressed my concerns and the manuscript is much clearer and improved. The logic of the experiments is more evident and appreciated by the reader.

A version with all changes highlighted would be useful.

Reviewer #2 (Remarks to the Author):

1.Lines 304-306, 354-358:

The description of the picture is problematic. For example, the amino acid residues are not clearly indicated in the figure.

2.Why do part of the work on SsCOMT need to be done? There is no clear explanation in the discussion section.

Reviewer #3 (Remarks to the Author):

the authors took all comments into account and revised the manuscript accordingly. I think that the manuscript is now in a condition to be published.

REVIEWERS' COMMENTS

Reviewer #1 (Remarks to the Author):

The authors have addressed my concerns and the manuscript is much clearer and improved. The logic of the experiments is more evident and appreciated by the reader.

A version with all changes highlighted would be useful.

Responses: We greatly appreciate your kind comments and we have now included a copy of the main text with all changes highlighted in a PDF file (named as Main text changes highlighted).

Reviewer #2 (Remarks to the Author):

1.Lines 304-306, 354-358:

The description of the picture is problematic. For example, the amino acid residues are not clearly indicated in the figure.

Responses: Thank you for your comment. We have added in color-coded labels for each amino acid in Fig. 5f.

2.Why do part of the work on SsCOMT need to be done? There is no clear explanation in the discussion section.

Responses: Thank you for your comment. We believe that characterizations of SsCOMT are needed to support its putative role in isorhapontigenin biosynthesis in wounded wild sugarcane stalks. Our data demonstrated that SsCOMT shares similar catalytic properties with SbCOMT, in particular, its catalytic regioselectivity towards stilbene B-ring, which favours the *O*-methylation of piceatannol at its B-ring over its A-ring, corroborating with the *in planta* stilbene profiles. qRT-PCR and semi-qRT-PCR further showed its constitutive expression in wounded wild sugarcane stalks. In our first revision, we did address the rationale of characterizing SsCOMT in the discussion (now in Lines 496-500), quoted as follows: "On the other hand, the presence of isorhapontigenin (B-ring *O*-methylated) is likely resulting from canonical activities of SsCOMT considering its constitutive expression in wounded wild sugarcane stalk (Supplementary Fig. 13b & 13d) and apparent catalytic regioselectivity towards stilbene B-ring (Fig. 7c), although genetic evidence is required for confirmation."

Reviewer #3 (Remarks to the Author):

the authors took all comments into account and revised the manuscript accordingly. I think that the manuscript is now in a condition to be published.

Responses: We greatly appreciate your kind comments.